



# A high-resolution (0.05°) global seamless continuity record (2002–2023) of near-surface soil freeze-thaw states via passive microwave and optical satellite data

Defeng Feng[1,2], Tianjie Zhao[1,2], Jingyao Zheng[1], Yu Bai[1], Youhua Ran[2,3], Xiaokang Kou[4,5], Lingmei Jiang[6], Ziqian Zhang[6], Pei Yu[7], Jinbiao Zhu[8,9,10], Jie Pan[8,9,10], Jiancheng Shi[11], Yuei-An Liou[12]

[1]National Key Laboratory of Remote Sensing and Digital Earth, Aerospace Information Research Institute, Chinese Academy of Sciences, Beijing 100101, China
[2]University of Chinese Academy of Science, Beijing 100049, China
[3]State Key Laboratory of Cryospheric Science and Frozen Soil Engineering, Heihe Remote Sensing Experimental Research Station, Northwest Institute of Eco-Environment and Resources, Chinese Academy of Sciences, Lanzhou 730000, China
[4]School of Civil Engineering, Shijiazhuang Tiedao University, Shijiazhuang 050043, China
[5]Key Laboratory of Roads and Railway Engineering Safety Control, Shijiazhuang Tiedao University, Ministry of Education, Shijiazhuang 050043, China
[6]National Key Laboratory of Remote Sensing and Digital Earth, Jointly Sponsored by Beijing Normal University and Aerospace Information Research Institute of Chinese Academy of Sciences, Faculty of Geographical Science, Beijing Normal University, Beijing 100875, China
[7]School of Geospatial Engineering and Science, Sun Yat-sen University, Zhuhai 519082, China
[8]Aerospace Information Research Institute, Chinese Academy of Sciences, Beijing 100094, China
[9]Aeronautical Remote Sensing Center, Chinese Academy of Sciences, Beijing 100094, China
[10]School of Electronic, Electrical and Communication Engineering, University of Chinese Academy of Sciences, Beijing 101408, China
[11]National Space Science Center, Chinese Academy of Sciences, Beijing 100190, China
[12]Hydrology Remote Sensing Laboratory, Center for Space and Remote Sensing Research, National Central University, Taoyuan 320317, China

*Correspondence to*: Tianjie Zhao (zhaotj@aircas.ac.cn)

**Abstract.** The global near-surface soil freeze-thaw (FT) states are crucial for understanding complex interactions with hydrological, ecological, and climatic processes. However, current remote sensing of FT states primarily relies on passive microwave remote sensing, which, despite its all-weather monitoring capabilities, suffers from low spatial resolution. This limitation restricts its application to hydroclimatological scales, precluding its use in finer-scale studies such as soil erosion and hydrometeorological applications. To address this, this study introduces a novel downscaling approach that integrates passive microwave and optical satellite data to generate a long-term (2002–2023), high-resolution (0.05°) dataset of global near-surface FT states, ensuring daily seamless continuity. The dataset was validated against in situ measurements, demonstrating that the high-resolution product maintains an overall accuracy of 83.78%, consistent with the coarse-resolution microwave-based dataset, while offering enhanced spatial detail. Comprehensive global trend analyses provided new insights into the dynamics of FT cycles, revealing that the average annual number of frost days in regions north of 45°N is 187.8 ± 12.7 days, with 14.35% of the area showing a decreasing trend in frozen persistence. Additionally, the average annual number of freeze onset dates is 240.3 ± 7.2, and 9.10% of the area exhibits a trend of delayed freeze onset. The high-resolution record



enables accurately monitoring FT states and providing detailed information, for a refined understanding of hydrological and ecological effects globally. The global 0.05° near-surface soil FT state dataset is freely available at

https://doi.org/10.11888/Cryos.tpdc.301551 (Zhao et al., 2024b).

## 1 Introduction

Frozen ground, including permafrost and seasonally frozen ground, covers nearly 66 million km², accounting for 52.5% of the total global land surface area (Kim et al., 2011; McDonald and Kimball, 2006). Most of these regions experience seasonal freeze-thaw (FT) state cycles, which refer to the phase transition between water and ice in the soil pores of surface layers

(Zhang et al., 2010). These FT state cycles significantly impact on surface runoff, energy balance, and carbon cycling (Kimball et al., 2004; Wang et al., 2024), thereby influencing climate (Peng et al., 2016; Poutou et al., 2004), hydrological (Gouttevin et al., 2012; Gray et al., 1985), ecological (Black et al., 2000) and biogeochemical processes (Panneer Selvam et al., 2016; Schaefer et al., 2011; Xu et al., 2013). Therefore, the dynamics of surface soil FT states are recognized as a significant indicator of global climate warming, with broad implications for the Earth's environment and human society (Beer et al., 2018; Yang et

al., 2013).

Over the past few decades, numerous studies have focused on detecting soil FT status. Although many traditional methods based on in situ observations offer advantages in terms of accuracy and reliability (Wei et al., 2011; Cary et al., 1979; Zhang et al., 2007), their labor-intensive nature, coupled with sparse spatial coverage and limited representativeness, restricts their ability to meet the comprehensive requirements of frozen ground research. In contrast, satellite remote sensing offers an

effective and rapid alternative, providing extensive spatial and temporal coverage for monitoring FT status.

Due to the critical importance of monitoring global surface FT states in hydroclimatology research, FT has been identified as an Essential Climate Variable (ECV) by the Global Climate Observation System (GCOS) (Bojinski et al., 2014). FT cycles occur frequently and are highly spatially heterogeneous due to variations in topography, vegetation, soil properties, and snow characteristics (Chang et al., 2015; Jiang et al., 2020). As a result, the GCOS has specified explicit requirements for FT

estimation. While an ideal spatial resolution of 1 km and a temporal resolution of 6 hours are recommended, the application of FT monitoring can be improved if FT records achieve a spatial resolution of 10 km and daily temporal resolution. However, current remote sensing techniques are unable to directly meet these requirements for monitoring FT dynamics.

Satellite microwave remote sensing methods are less susceptible to potential degradation from solar illumination effects and atmospheric cloud/aerosol contamination, while remaining sensitive to changes in landscape dielectric properties due to liquid

water content variations in soils. This characteristic makes them highly applicable in consistent FT detection (England, 1990; Kou et al., 2017; McDonald et al., 2004b; Wu et al., 2022; Zhao et al., 2014). Passive microwave remote sensing is an effective technique for monitoring global surface FT processes owing to radiometers' shorter revisit intervals and larger coverage areas (Kim et al., 2011; Kou et al., 2017; Zuerndorfer and England, 1992). Additionally, it can detect microwave radiation from specific depths within the surface soil and is sensitive to changes in soil dielectric properties, thereby enhancing its applicability



in FT detection (McDonald et al., 2004b). In recent years, with the development of new-generation passive microwave radiometers, various corresponding FT discrimination algorithms have been constructed, including the dual-index algorithm (Han et al., 2015; Judge et al., 1996; Zuerndorfer and England, 1992; Zuerndorfer et al., 1990), the decision tree algorithm (Jin et al., 2009), the discriminant function algorithm (DFA) (Hu et al., 2019; Kou et al., 2017, 2018; Zhao et al., 2011), the seasonal threshold method (Kim et al., 2011) and the polarization ratio(PR)-based algorithm (Rautiainen et al., 2016; Roy et al., 2015).

However, the relatively coarse spatial resolution of current passive microwave radiometers limits the retrieval of high-resolution information (Chai et al., 2014; Han et al., 2015; Zhao et al., 2011; Zhou et al., 2016). Moreover, spatial heterogeneity, often caused by mixed pixels resulting from the coarse spatial resolution, introduces uncertainty into FT data derived from passive microwave remote sensing.

Optical remote sensing techniques provide high spatial resolution information, such as land surface temperature (LST), which

can be used to infer surface FT states. For example, the Moderate Resolution Imaging Spectroradiometer (MODIS) on Aqua and Terra satellites provides LST products with high accuracy, potentially allowing for monitoring global FT states. However, LST primarily reflects the linear changes in temperature during the FT process and cannot capture the abrupt changes in dielectric properties that occur between the frozen and thawed soils. In addition, these products are significantly affected by discontinuities in coverage due to cloud contamination, vegetation, and snow cover (Cary et al., 1979; Langer et al., 2013;

Chen et al., 2021; Running, 1998). Recent research focus is exemplified by efforts to generate seamless datasets (Li et al., 2018; Yao et al., 2023; Yu et al., 2022; Zhang et al., 2022, 2020; Zhao et al., 2024a).

Active microwave sensors, divided into radars and scatterometers, receive backscattering coefficients as the echo signals, which are primarily related to soil structure and dielectric properties. Active microwave remote sensing, particularly through synthetic aperture radars (SARs), provides observations of landscape FT states at resolutions on the kilometer scale or finer.

Consequently, many active microwave FT discrimination algorithms have been developed, including the FT threshold value method (Du et al., 2015; Kim et al., 2012; Kimball et al., 2004; Way et al., 1997), a change detection algorithm (Frolking et al., 1999), and an edge detection algorithm (Canny, 1986; McDonald et al., 2004a). However, most satellite-based SARs have longer revisit periods than passive microwave radiometers, preventing them from meeting the temporal resolution requirements of FT monitoring.

Existing methods fall short in fulfilling the scientific requirements for high-resolution detection of FT states and are predominantly dependent on microwave observations. To overcome these limitations, downscaling techniques based on multi-source data fusion have been proposed (Giorgi and Mearns, 1991; Hanssen-Bauer et al., 2005; Xu et al., 2019). Among these, statistical downscaling methods (Bierkens et al., 2000; Vaittinada Ayar et al., 2016) are commonly applied in high-resolution monitoring studies (Fan et al., 2005; Hertig and Jacobeit, 2008), which quickly establish statistical or empirical relationships

between downscaling predictors and predictands, thereby enabling the reconstruction of coarse-resolution data at finer scales. This has inspired researchers to integrate multi-source remote sensing data, leveraging the high sensitivity of passive microwave signals to FT dynamics and the finer spatial details of optical datasets (Zhao et al., 2017).





The detection of frozen and thawed soil can rely on two critical characteristics: temperatures below 0 °C and the presence of ice. Therefore, remote sensing observations can be utilized to infer these conditions by monitoring the physical temperature

and liquid water content. Microwave and optical remote sensing each offer distinct advantages in this context. Microwave observations are extensively employed to detect FT dynamics through various discrimination algorithms. In particular, the DFA has proven to be highly effective in generating consistent, long-term, daily global FT state products. This algorithm, which utilizes brightness temperature (TB) data from the Advanced Microwave Scanning Radiometer for EOS (AMSR-E) and its successor, the Advanced Microwave Scanning Radiometer 2 (AMSR2), has demonstrated both high validation accuracy

and reasonable consistency (Hu et al., 2019; Wang et al., 2019a). One of the key discriminant indicators in this algorithm is TB at 36.5 GHz due to its strong correlation with LST. The other important indicator is the quasi-emissivity (Qe), the ratio between TB at 18.7 GHz and TB at 36.5 GHz, which captures the variations in landscape dielectric properties (ice/water content) across different FT conditions (Zhang et al., 2010; Zhao et al., 2011). Despite its strengths in physical mechanisms, the coarse spatial resolution of passive microwave sensors constrains its application in detailed monitoring. In comparison,

optical remote sensing provides data with high spatial resolution, offering an alternative for detecting detailed FT states. A previous study attempted to generate high-resolution FT maps, but the optical data used were restricted to LST and did not incorporate parameters characterizing soil ice/water content. (Hu et al., 2017). The apparent thermal inertia (ATI), derived from optical data, has been shown to effectively monitor soil moisture conditions (Qin et al., 2013; Song and Jia, 2016; Van Doninck et al., 2011; Veroustraete et al., 2012; Verstraeten et al., 2006). Thus, integrating ATI into data fusion might further

enhance the ability to generate high-resolution FT records (Liang et al., 2024; Yao et al., 2023; Yu et al., 2022; Zhang et al., 2022).

The objective of this study is to enhance the spatial resolution of FT detection products without compromising the accuracy of passive microwave-based FT products, as derived from AMSR-E/2 TB data through the DFA. To achieve this enhancement, the study employs downscaling indicators, specifically the MODIS-based LST and ATI, which serve to encapsulate soil

moisture information. Both the original coarse-resolution and the resultant high-resolution FT records were validated by in situ soil temperatures, thereby facilitating an assessment of accuracy variations post downscaling. Subsequent trend analyses of the high-resolution FT records were conducted to reflect the detailed dynamics of FT states. The resultant downscaled, high-resolution FT states align with the expanding spatial and temporal resolution requirements of GCOS for FT monitoring, thereby providing a valuable tool in cryospheric and ecological studies.

**2 Data**

**2.1 AMSR-E and AMSR2 brightness temperature**

The near-surface FT downscaling method integrates data from passive microwave and optical remote sensing. TB observations from passive microwave sensors were obtained from AMSR-E and its successor, AMSR2. AMSR-E, which was carried on NASA's Aqua satellite, operated from May 2002 to October 2011 and provided six microwave bands (6.9, 10.65, 18.7, 23.8,



36.5, and 89 GHz), each available in both horizontal (H) and vertical (V) polarization (Cho et al., 2017). AMSR2, launched in May 2012 and mounted on Japan Aerospace Exploration Agency's (JAXA's) Global Change Observation Mission-Water 1 (GCOM-W1) satellite, retains most of AMSR-E's physical properties, except for a larger antenna reflector and an extra C-band channel (7.3 GHz) (Cho et al., 2017). Both sensors share the same spatial resolution of 0.25° and provide measurements at 13:30 (ascending) and 1:30 (descending) local time at the equator. This study utilized TB data from the 18.7H and 36.5V

channels of the AMSR-E/2 Level-3 TB standard product to discriminate near-surface FT states at a 0.25° grid resolution. All available TB data from 2002 to 2023 were utilized, excluding the missing observations between AMSR-E and AMSR-2.

## 2.2 Global spatiotemporally continuous MODIS LST dataset

The spatiotemporally continuous MODIS LST dataset was utilized in this study, derived from two products: MODIS/Terra LST Daily L3 Global 0.05° CMG (MOD11C1) and MODIS/Aqua LST Daily L3 Global 0.05° CMG (MYD11C1). These

products are separately obtained from the Terra and Aqua satellites, both of which cross the equator at the local time of 13:30 and 01:30 during ascending and descending orbits, respectively. However, cloud contamination introduces data gaps in these products. To address this issue, a data interpolation and reconstruction method was applied, enabling the generation of spatiotemporally continuous LST records. Furthermore, the clear-sky LSTs were corrected to all-weather LSTs, enabling the retrieval of more realistic information (Yu et al., 2022; Zhao and Yu, 2021). These datasets under clear-sky and all-weather

conditions exhibit satisfactory accuracy and are therefore suitable for high-resolution optical remote sensing inputs in the development of the downscaling algorithm.

## 2.3 GLASS Albedo

Spatiotemporally continuous land surface albedo products are essential for estimating global ATI, as outlined in the Introduction. In this study, we used the GLASS02B06 dataset, which is part of the Global Land Surface Satellite (GLASS)

project. The GLASS albedo series demonstrates accuracy comparable to that of the MODIS MCD43 albedo product, validated against FLUXNET site observations (Liu et al., 2013). Furthermore, GLASS albedo products address spatial gaps arising from cloud contamination and snow cover, thereby ensuring seamless and continuous products. This study employed the long-term clear-sky albedo dataset from GLASS02B06, which captures surface albedo at a 0.05° spatial resolution and provides new data every 8 days.

## 2.4 Land cover maps

In this study, the ancillary land cover maps were derived from the MODIS Land Cover Type CMG Yearly L3 Global 0.05° (MCD12C1) dataset (Friedl and Sulla-Menashe, 2022). With a spatial resolution of 0.05°, this dataset aligns with the resolution of the optical datasets mentioned earlier. It follows the 17-class International Geosphere-Biosphere Programme (IGBP) classification framework, as summarized in Table 1. The land cover dataset, illustrated in Fig. 1, was specifically utilized to



identify snow- and ice-covered areas as well as urban and built-up regions. Furthermore, the corresponding type percentage dataset was used to filter out pixels dominated by large water bodies, which were then explicitly marked in the FT data record.

**Table 1 IGBP land cover classification framework.**

| value | IGBP Classes Abbreviated name | Full name | value | IGBP Classes Abbreviated name | Full name |
|---|---|---|---|---|---|
| 0 | WAT | Water Bodies | 9 | SAV | Savannas |
| 1 | ENF | Evergreen Needleleaf Forest | 10 | GRL | Grasslands |
| 2 | EBF | Evergreen Broadleaf Forest | 11 | PWL | Permanent Wetlands |
| 3 | DNF | Deciduous Needleleaf Forest | 12 | CRL | Croplands |
| 4 | DBF | Deciduous Broadleaf Forest | 13 | URB | Urban and Built-Up |
| 5 | MXF | Mixed Forest | 14 | CRM | Cropland Mosaics |
| 6 | CSH | Closed Shrubland | 15 | SNI | Snow and Ice |
| 7 | OSH | Open Shrubland | 16 | BSV | Barren/Sparsely Vegetated |
| 8 | WSA | Woody Savannas | | | |

## 2.5 In situ soil temperature

Validation of the downscaled FT dataset was conducted using soil temperature measurements obtained from 41 dense observation networks and three sparse networks (SCAN, SNOTEL, USCRN). Among these, 42 networks were provided by the International Soil Moisture Network (ISMN), a global collaboration that delivers in situ measurements of soil moisture and related variables (Dorigo et al., 2013, 2021). Additionally, two networks (Naqu and Pali) were obtained from the Tibetan Plateau Observatory (Tibet-Obs), which focuses on plateau-scale monitoring of soil moisture and temperature (Su et al., 2011; Zhang et al., 2021a, b). Although the sampling time points vary across networks, the measurements are consistently conducted at hourly intervals. This study selected long-term in situ soil temperature data from 1,027 stations within 44 global networks at a depth of 0–5 cm. Fig. 1 illustrates the spatial distribution of these stations.



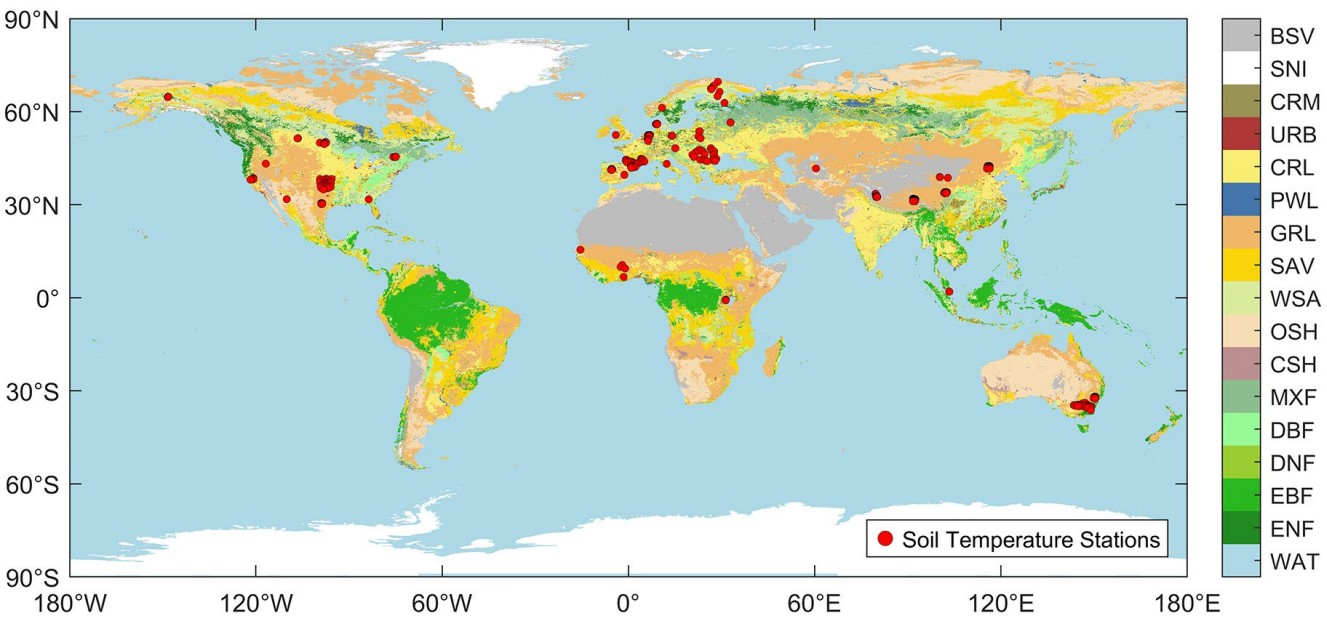

**Figure 1: ISBP global land cover map for 2019, depicting the spatial distribution of ground measurement sites within the ISMN and Tibet-Obs networks.**

## 2.6 Data pre-processing

The inter-calibration of different satellite instruments is for establishing consistent data records of Earth's environment (Chander et al., 2013). Although AMSR-E and AMSR2 share numerous similarities, calibrating these two sensors is necessary due to a 5 K difference in observed TB caused by different calibration procedures and other issues (Okuyama and Imaoka, 2015). Hu et al. (2019) introduced an inter-calibration linear model utilizing overlapping TB observations and the least squares method. This inter-calibration model was applied to TB data at the 18.7 GHz and 36.5 GHz bands, generating a long-term passive microwave TB dataset. The inter-calibration model equations are as follows:

$$TB_{\text{AMSRE\_18.7H}} = 1.0189 \times TB_{\text{AMSR2\_18.7H}} - 5.2717 \,, \tag{1}$$

$$TB_{\text{AMSRE\_18.7V}} = 1.0577 \times TB_{\text{AMSR2\_18.7V}} - 16.2042 \,, \tag{2}$$

$$TB_{\text{AMSRE\_36.5H}} = 1.0073 \times TB_{\text{AMSR2\_36.5H}} - 4.7723 \,, \tag{3}$$

$$TB_{\text{AMSRE\_36.5V}} = 1.0135 \times TB_{\text{AMSR2\_36.5V}} - 6.3914 \,, \tag{4}$$

where $TB_{\text{AMSRE\_18.7H}}$ and $TB_{\text{AMSR2\_18.7H}}$ denote TB at 18.7 GHz in horizontal polarization, obtained from AMSR-E and AMSR2, respectively. $TB_{\text{AMSRE\_18.7V}}$ and $TB_{\text{AMSR2\_18.7V}}$ correspond to vertical polarization at the same frequency. Other terms in the equations follow the same conventions.



To meet the requirements of daily ATI estimation, a linear interpolation method was employed to transform the GLASS 8-day
shortwave clear-sky albedo datasets into daily data (Sohrabinia et al., 2014).

## 3. Methodology

### 3.1 FT discrimination from passive microwave observations

The DFA is a surface FT discrimination method developed using AMSR-E and AMSR2 data, demonstrating high accuracy

compared to existing FT products. Near-surface FT variations are closely associated with soil temperature and moisture, which
are reflected by the TB at 36.5 GHz in vertical polarization ($TB_{36.5V}$) and the "Quasi-emissivity" (Qe). Qe is defined as the
ratio of the TB at 18.7 GHz in horizontal polarization ($TB_{18.7H}$) to $TB_{36.5V}$, serving as a representative measure of soil water
content. Therefore, $TB_{36.5V}$ and Qe were selected as key parameters for FT discrimination (Zhao et al., 2011). Additionally,
the DFA is parameterized to separately detect FT status during ascending and descending orbits (Wang et al., 2019b), as

expressed by the following equations:

$$FTI_A = -0.123 \times TB_{36.5V} + 11.842 \times Qe + 20.650 \,, \tag{5}$$

$$FTI_D = -0.209 \times TB_{36.5V} + 9.384 \times Qe + 43.697 \,, \tag{6}$$

$$Qe = \frac{TB_{18.7H}}{TB_{36.5V}} \,, \tag{7}$$

where $FTI_A$ and $FTI_D$ represent the FT status for ascending and descending orbits, respectively. Through these equations, a

long-term 0.25° FT record was generated using AMSR-E and AMSR2 TB data.

### 3.2 Estimation of apparent thermal inertia

ATI, as a substitute indicator for thermal inertia (TI), has proven effective in monitoring soil moisture conditions, thereby
facilitating the downscaling of coarse-resolution FT states (Qin et al., 2013; Song and Jia, 2016; Van Doninck et al., 2011;
Veroustraete et al., 2012; Verstraeten et al., 2006). ATI quantifies the temperature rise due to the Earth's surface absorbing

radiant energy. Given that water has a relatively large heat capacity, higher soil water content provides greater resistance to
external thermal fluctuations (Qin et al., 2013).

In this study, ATI is introduced as an indicato of soil moisture and is calculated as follows:

$$ATI = C \frac{1-a_0}{DTA} \,, \tag{8}$$

where $a_0$ is the actual surface albedo, derived from interpolated daily surface albedo data. $C$ denotes the solar correction factor,

which accounts for spatial and temporal variations in solar flux due to latitude and solar declination. The diurnal LST cycle's
amplitude, denoted as $DTA$, represents the largest variation in LST observed within a single day.

$C$ is calculated as follows:





$$C = \sin \varphi \sin \delta \, (1 - \tan^2 \varphi \tan^2 \delta)^{1/2} + \cos \varphi \cos \delta \arccos(- \tan \varphi \tan \delta) \,, \tag{9}$$

where $\varphi$ is the latitude and $\delta$ represents the solar declination, calculated by:

$$\delta = 0.006918 - 0.399912 \cos(\Gamma) + 0.070257 \sin(\Gamma) - 0.006758 \cos(2\Gamma) + 0.000907 \sin(2\Gamma) - 0.002697 \cos(3\Gamma) + 0.00148 \sin(3\Gamma) \,, \tag{10}$$

where $\Gamma$ represents the day angle, given by:

$$\Gamma = \frac{2\pi(n_d - 1)}{365.25} \,, \tag{11}$$

and $n_d$ is the day number of the year.

The daily LST amplitude $DTA$ in Eq. (8) can be calculated as:

$$\frac{DTA}{2} = \frac{n \sum_{i=1}^{n} \cos(\omega t_i - \psi) T_i - n \sum_{i=1}^{n} \cos(\omega t_i - \psi) \sum_{i=1}^{n} T_i}{n \sum_{i=1}^{n} \cos(\omega t_i - \psi) - \left(\sum_{i=1}^{n} \cos(\omega t_i - \psi)\right)^2} \,, \tag{12}$$

where $T_i$ is the LST recorded at time $t_i$, $\omega$ is the angular velocity of the Earth's rotation. The parameter $n$ represents the number of LST observations for one pixel in a day ($n \equiv 4$ in this study), and $\psi$ is the phase angle, calculated as:

$$\psi = \arctan(\xi) + \pi \,, \tag{13}$$

$$\xi = \frac{(T_1 - T_3)(\cos(\omega t_2) - \cos(\omega t_4)) - (T_2 - T_4)(\cos(\omega t_1) - \cos(\omega t_3))}{(T_2 - T_4)(\sin(\omega t_1) - \sin(\omega t_3)) - (T_1 - T_3)(\sin(\omega t_2) - \sin(\omega t_4))} \,, \tag{14}$$

where Eq. (14) is used to calculate the diurnal LST cycle and requires clear-sky LST observations at four specific times during the day: Aqua/night (01:30), Terra/day (10:30), Aqua/day (13:30), and Terra/night (22:30). These observations are derived from the spatiotemporally continuous MODIS clear-sky LST products. The method of ATI calculation is adapted from Van Doninck et al. (2011). Consequently, a long-term 0.05° daily ATI dataset has been prepared for downscaling FT status.

## 3.3 Spatial downscaling of FT status

The FTI is characterized by its quantitative form, expressed in decimal values instead of binary values (Zhao et al., 2017). This feature enables its integration with other satellite-derived parameters, thereby enhancing its utility in remote sensing applications. As defined by soil temperature and emissivity from microwave data in Eqs. (5) and (6), the FTI exhibits a strong correlation with LST, thereby demonstrating its relevance in thermal analysis. (Zhao et al., 2017). Moreover, the emissivity properties of soil are susceptible to changes in soil water content, which can be captured through ATI estimation.

Consequently, the assumed linear relationship between microwave and optical datasets is mathematically expressed as follows:

$$FTI = a \cdot LST + b \cdot ATI + c \,, \tag{15}$$

where the coefficients $a$, $b$ and $c$ are determined through linear regression analysis. The MODIS/Aqua LST dataset was selected to ensure temporal consistency with the AMSR-E TB product.



Given the annual FT cycles, the downscaling approach was applied to the data on a yearly basis, following four steps as illustrated in Fig. 2.

The 0.05° LST and ATI data were resampled to align with the lower 0.25° resolution of the microwave-derived FTI. This resampling process involved averaging the data across 5×5 grid cells.

The FTI, LST, and ATI datasets were divided into yearly vectors, and the data within each vector were arranged in
chronological order.

Linear regression analysis was performed on these three data vectors of each pixel, resulting in six coefficient matrices for $a$, $b$, and $c$ at the ascending and descending times:

$$FTI_{\text{LR}} = a_{\text{LR}} \cdot LST_{\text{LR}} + b_{\text{LR}} \cdot ATI_{\text{LR}} + c_{\text{LR}} ,\qquad(16)$$

where the abbreviations "HR" and "LR" refer to high and low resolution, respectively. Observational data corresponding to
invalid FTI values were excluded from the regression analysis.

Once the coefficients $a_{\text{LR}}$, $b_{\text{LR}}$, and $c_{\text{LR}}$ were expanded into 5×5 grids at a spatial resolution of 0.05°, they were applied in a regression model with the high-resolution (0.05°) LST and ATI data to generate the downscaled FTI values:

$$FTI_{\text{HR}} = a_{\text{LR}} \cdot LST_{\text{HR}} + b_{\text{LR}} \cdot ATI_{\text{HR}} + c_{\text{LR}} ,\qquad(17)$$

This process produced a finer spatial resolution compared to the original 0.25° FT maps derived from microwave data.
Additionally, the land cover product was utilized to identify areas characterized by permanent snow and ice, substantial water cover, and urban or built-up lands, ensuring an accurate representation in the downscaled dataset.



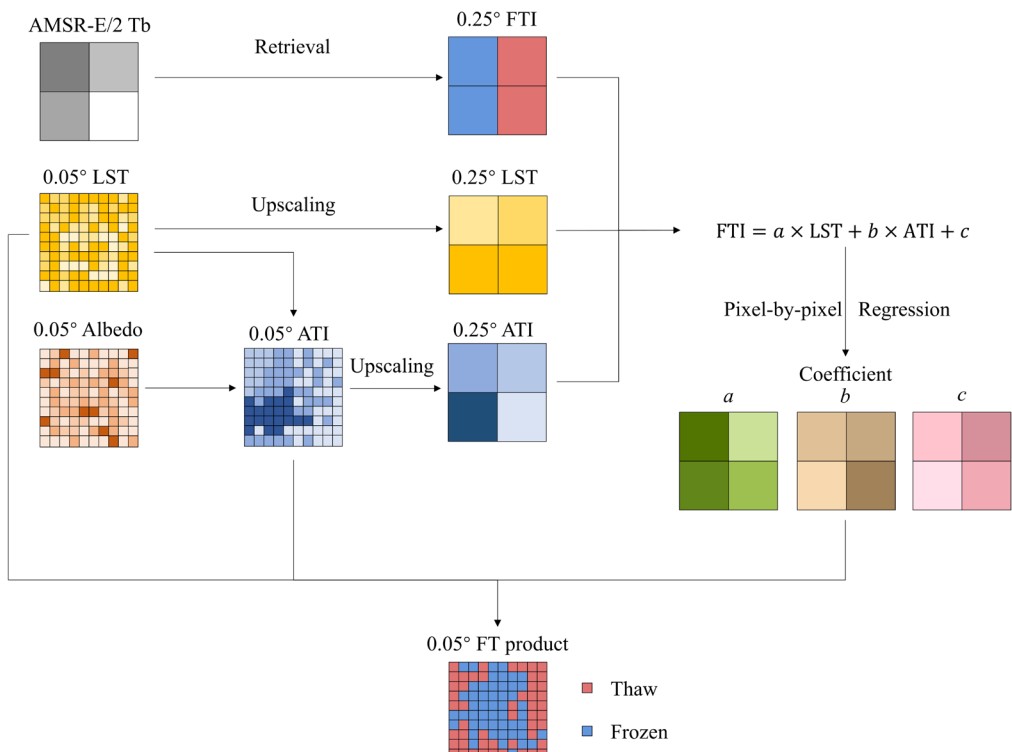

**Figure 2: Schematic diagram of spatial downscaling methodology.**

### 3.4 Validation

Validating the downscaled FT product is essential to ensure its accuracy and reliability. In this study, soil temperature data from ISMN and Tibet-Obs ground stations were used to validate the downscaled FT results. Since satellite observations are divided into ascending and descending orbit data, the FT records were validated independently for each orbit. Frozen and thawed states in in situ soil temperature data are classified as follows:

$$\begin{cases} T \leq 0, \text{frozen} \\ T > 0, \text{thawed} \end{cases} \text{'} \tag{18}$$

To validate satellite data against ground measurements, the in situ observations must first be selected and pre-processed. In situ observations temporally closest to the satellite observation times are selected. ISMN observation data are recorded at hourly intervals, while the MODIS LST product provides observations at 20-minute increments. Thus, in situ data recorded at the hour closest to the satellite observation time are used for validating FT classification. Missing data are excluded from the validation process. Due to the properties of microwave sensors, data gaps may occur in the 0.25° FT product, where the

corresponding in situ observations are not included in the accuracy validation. Furthermore, only ISMN data labeled as "Good" quality is selected for validation.



Accuracy was calculated separately for the 0.25° and 0.05° FT products. The validation process, illustrated in Fig. 3, was conducted using a single, larger 0.25° grid cell as an example.

FT conditions derived from in situ observations were considered the actual results. For a grid cell with a resolution of 0.25°,
the average of all in situ soil temperature observations within the cell was calculated. The grid cell was subsequently classified according to the criteria delineated in Eq. (18), distinguishing between frozen and thawed states.

The FT classification results for the larger grid cell from both products were validated using in situ data. For the 0.25° product, the FT classification of the grid cell was directly compared to the in situ observation results. For the 0.05° product, each 0.25° grid cell contains 25 pixels. If the majority of these pixels (i.e., more than 13) were classified as frozen, the entire grid cell was
labeled as frozen in the 0.05° product; otherwise, it is labeled as thawed.

The classification accuracies of the two products were calculated. Within each grid cell, the number of true and false FT classifications over the entire time series was counted to determine the discrimination accuracy using the following formula:

$$\text{Accuracy} = \frac{FF+TT}{FF+FT+TF+TT}, \tag{19}$$

where FF denotes the number of classifications where both the in situ observation and the satellite observation indicate a frozen
state, and FT denotes the number of classifications where the soil is frozen, but the satellite observation indicates thawed conditions. TF and TT are defined similarly.

The same validation procedures were applied to all other grid cells containing in situ observations. The discrimination accuracy before and after downscaling was calculated and compared to assess any changes in accuracy resulting from the downscaling process.





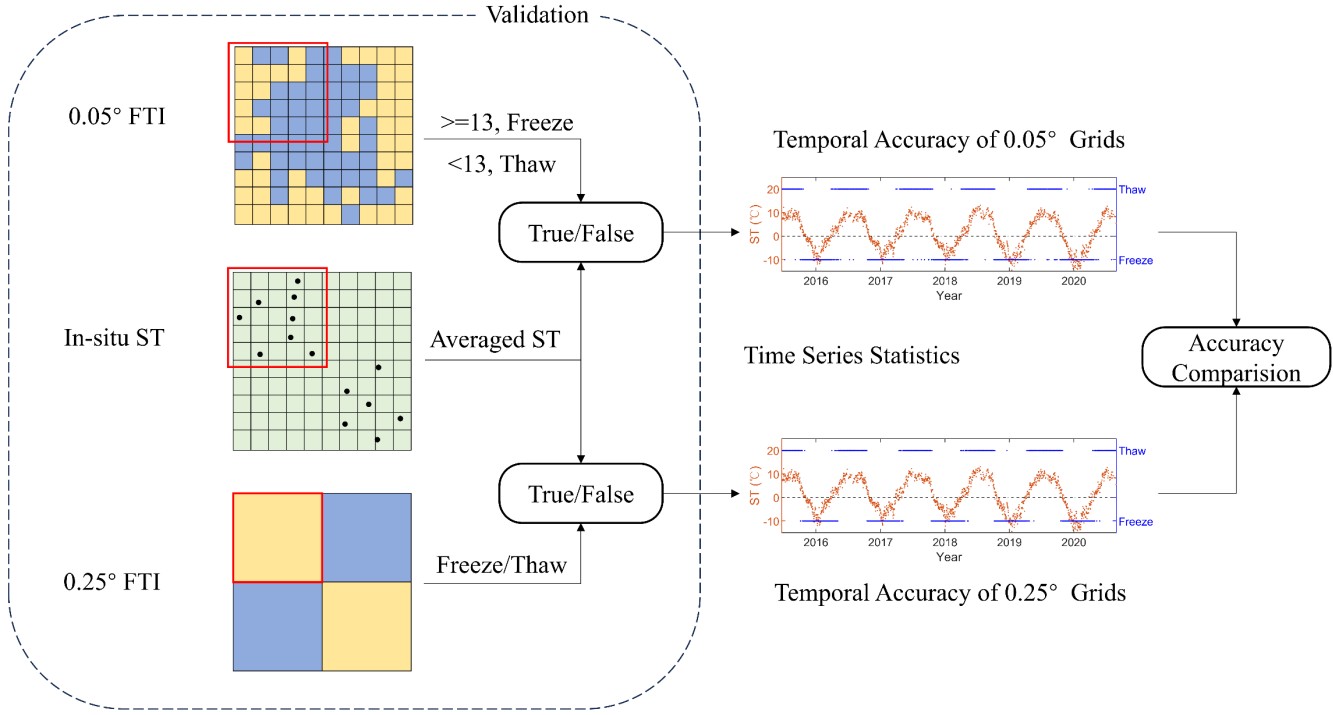

**Figure 3: Schematic diagram of the validation method for comparing the FT product before and after downscaling.**

### 3.5 Trend analysis

Trend analysis for global FT cycles is crucial for understanding temporal changes across different regions, which may indicate
impacts associated with climate change. To analyze the annual global FT trend based on the downscaled product, this study
employs two key parameters: the frost day and the frozen onset date.

A frost day is defind as the number of days within a year in which the lowest recorded temperature is less than 0 °C. The
minimum temperature can be inferred to be below 0 °C if freezing is detected at the satellite's descending time for the same
pixel. Therefore, in the satellite FT product, a frost day is identified as a day when the pixel is classified as frozen. In this study,
the calculation and analysis of frost days rely on the descending FT product. The freeze onset date refers to the first day a pixel
begins freezing and remains in a frozen state for more than two weeks. The initial day of this freezing period is designated as
the freeze onset date. Both the frost day and the freeze onset date offer significant insights into the global distribution of near-
surface FT variations. These two parameters are essential for comprehending the global temporal and spatial dynamics of
freezing events.

To further analyze the trends of frost days and freeze onset dates, time series data for both parameters were generated for
global-scale analysis using Sen's Slope and Mann-Kendall (MK) test.

The equation for Sen's Slope is defined as:





$$\beta = Median(\frac{X_j - X_i}{j-i}), \forall j > i \,, \tag{20}$$

where $X_j$ and $X_j$ represent the data values at times $i$ and $j$, respectively, and $Median(\cdot)$ represents the median calculation. This
equation computes the median of the slopes between all pairs of data points. A positive $\beta$ reflects an increasing trend, while a
negative $\beta$ indicates a decreasing trend.

The MK test is a statistical procedure that is employed to evaluate the null hypothesis that the data are independent and
identically distributed (Mann, 1945; Kendall, 1948). It is employed to detect trends in time series data without assuming a
linear relationship. Given a series $X_t$ of length $n$, the null hypothesis posits that the values in $X_t$ are independent. The MK test
statistic $S$ is calculated as:

$$S = \sum_{i=1}^{n-1} \sum_{j=i+1}^{n} \text{sgn}(X_j - X_i) \,, \tag{21}$$

where $n$ is the sample size, defined as the number of data points, and $\text{sgn}(X_j - X_i)$ is the sign function, defined as

$$\text{sgn}(X_j - X_i) = \begin{cases} 1, & \text{if } X_j > X_i \\ 0, & \text{if } X_j = X_i \,, \\ -1, & \text{if } X_j < X_i \end{cases} \tag{22}$$

The test statistic $S$ evaluates the cumulative number of positive differences that exceed negative differences. In this study,
since the frost day and freeze onset date of each year in the same pixel are unique, the variance of $S$ is computed as

$$Var(S) = \frac{n(n-1)(2n+5)}{18} \,, \tag{23}$$

If the sample size $n > 10$, the standard normal test statistic $Z$ is calculated as:

$$Z = \begin{cases} \frac{S-1}{\sqrt{Var(S)}}, & \text{if } S > 0 \\ 0, & \text{if } S = 0 \,, \\ \frac{S+1}{\sqrt{Var(S)}}, & \text{if } S < 0 \end{cases} \tag{24}$$

Trend testing is conducted at a specified significance level, $\alpha$. If the absolute value of $Z$ exceeds the critical value $Z_{1-a/2}$, the
null hypothesis of no trend is rejected. In this study, the test was performed at $\alpha = 0.05$, corresponding to a 95% confidence
interval. Under these conditions, if $|Z| > 1.96$, the null hypothesis is rejected. The results of the test are categorized into five
trend types, as summarized in Table 2. Sen's Slope quantifies the direction of temporal variation as positive or negative, while
the MK test evaluates the statistical significance of these trends. This combined approach allows for the assessment of both
the magnitude and significance of trends in frost days and freeze onset dates, providing insights into changes in global FT
cycles that may be linked to climate change.

**Table 2 Classification of trends based on the MK test**





| $\beta$ | Z | Trend features |
|---|---|---|
| **$\beta > 0$** | $\lvert Z \rvert \geq 1.96$ | Significant increase |
| | $0 < \lvert Z \rvert < 1.96$ | Slight increase |
| **$\beta = 0$** | $\lvert Z \rvert = 0$ | No trend |
| **$\beta < 0$** | $0 < \lvert Z \rvert < 1.96$ | Slight increase |
| | $\lvert Z \rvert \geq 1.96$ | Significant increase |

## 4 Results and Discussions

### 4.1 Result of downscaled FT product

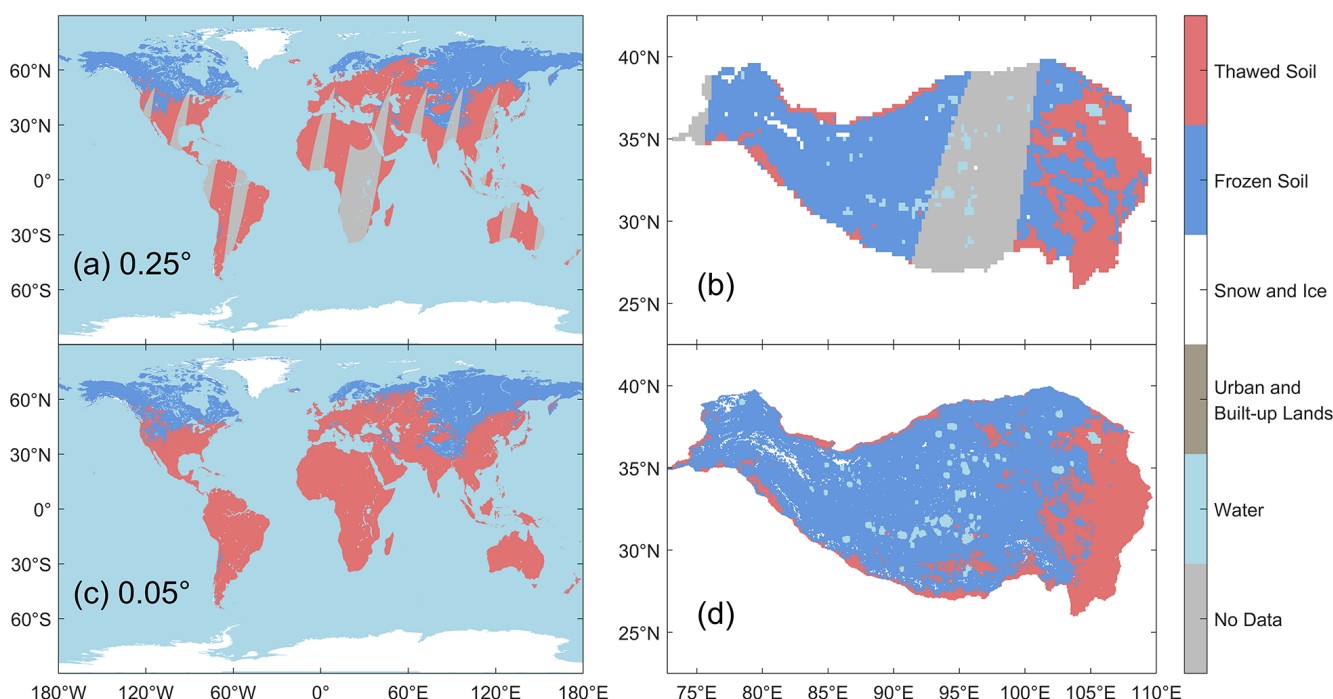

**Figure 4: Maps of the FT product for (a) the global scale and (b) the Qinghai-Tibet Plateau at 01:30 on April 1, 2019, derived from microwave data with a 0.25° spatial resolution. Additionally, (c) and (d) display the same conditions in the downscaled FT product at a higher resolution of 0.05°.**

According to the methodology described in Sect. 3.3, a global near-surface FT states dataset with a 0.05° resolution was generated, spanning the years 2002 to 2023. This dataset provides daily FT information for both daytime and nighttime, corresponding to the satellite's ascending (13:30) and descending (01:30) orbits, respectively. The left panel of Fig. 4 illustrates the global downscaled FT product for January 1, 2019, at 01:30, showing the FT status during the descending orbit. The dataset

classifies land surface conditions into the following categories: thawed soil, frozen soil, snow and ice, urban and built-up lands, water bodies, as well as regions with missing data, and water-influenced areas (labeled as "water" in Fig. 4).

An important point to consider is that areas near water bodies may be misclassified as frozen due to the high soil moisture content detected by microwave TB data. However, these areas are unlikely to freeze, particularly in coastal regions where temperatures consistently remain above 0°C. To improve the reliability of the FT product, these water-influenced areas should be carefully delineated and flagged.

The right panel of Fig. 4 presents a comparison of the FT product for the Qinghai-Tibet Plateau, highlighting the enhanced resolution of the 0.05° product in capturing FT states. The downscaled product more precisely classifies soil freezing and

thawing states, as well as specific surface features that are more difficult to detect in the 0.25° product. This demonstrates the enhanced capability of the 0.05° product to capture FT cycles at the local scale, compensating for the resolution limitations of the original 0.25° product.

This downscaled product, with its enhanced spatial resolution, allows for a more detailed analysis of global FT dynamics, providing a more reliable interpretation of global FT patterns. This capability is crucial for understanding FT cycles and their

associated environmental impacts.

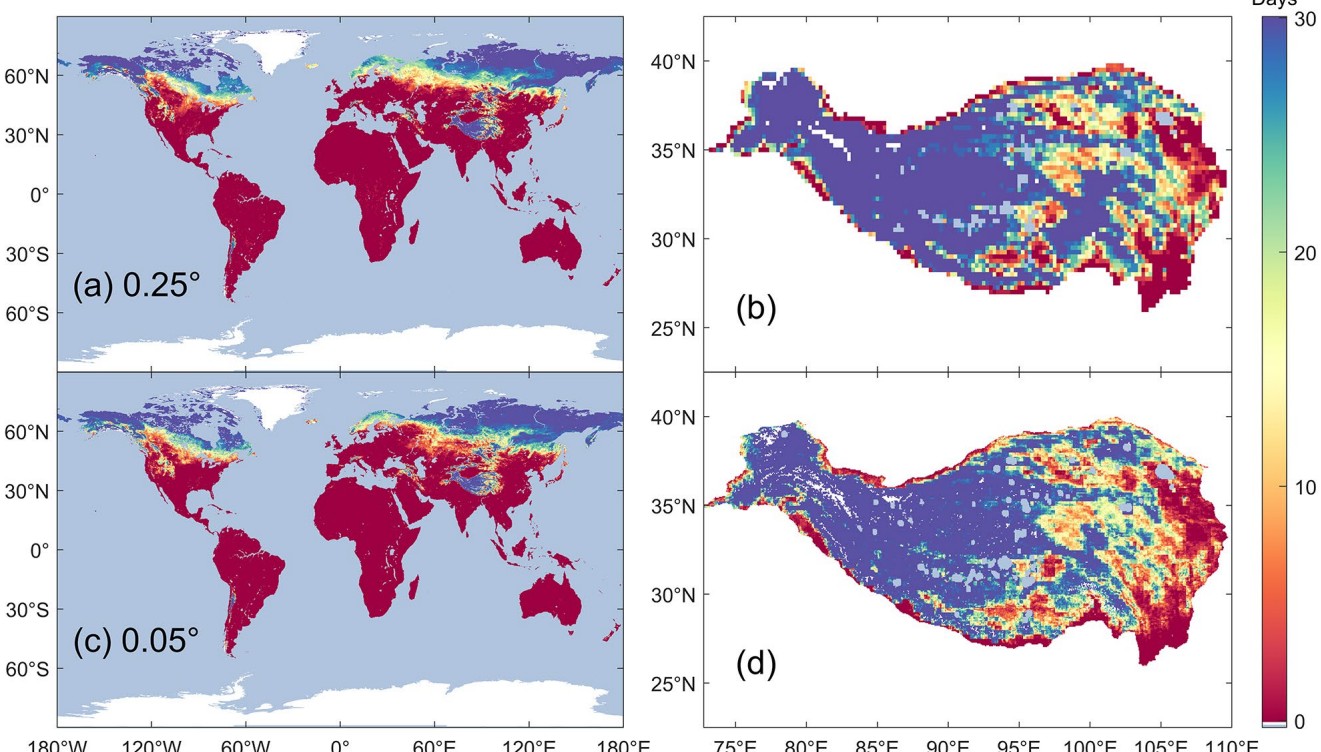

**Figure 5: Maps of frost days in April 2019 for (a) the global scale and (b) the Qinghai-Tibet Plateau, derived from microwave data with a 0.25° spatial resolution during the descending orbit time. Additionally, (c) and (d) show the same conditions in the downscaled descending FT product at a resolution of 0.05°.**

Fig. 5 illustrates frost days for April 2019 at two spatial resolutions, 0.25° and 0.05°, for both the global scale and the Qinghai-Tibet Plateau.

The left panel compares global frost days for a single month, showing that the 0.25° product exhibits a more gradual variation in frost days across the northern hemisphere. In contrast, the 0.05° product reveals more detailed and intricate patterns of frost day variation. For instance, in southern Russia and parts of North America, the 0.05° product captures more nuanced frost day

variations, highlighting significant differences in frost day counts between neighboring grid cells. Meanwhile, the 0.25° product displays relatively uniform changes on frost days, reflecting less variation due to its coarser resolution.

The right panel of Fig. 5 compares frost days for the Qinghai-Tibet Plateau, illustrating finer variations at the higher spatial resolution. This demonstrates that the downscaled product provides a more detailed distribution of frost days, while also capturing the trend of frost day variation at a finer spatial scale.

In summary, the downscaled FT product offers significantly more detailed information, especially in regions with complex FT dynamics. This improvement in spatial resolution is crucial for our research, as it enables a more refined understanding of global FT processes and their environmental impacts. The ability to capture finer details and regional variations is a key advantage of the downscaling approach, which effectively meets the primary objectives of this study.

## 4.2 Validation with in situ soil temperature

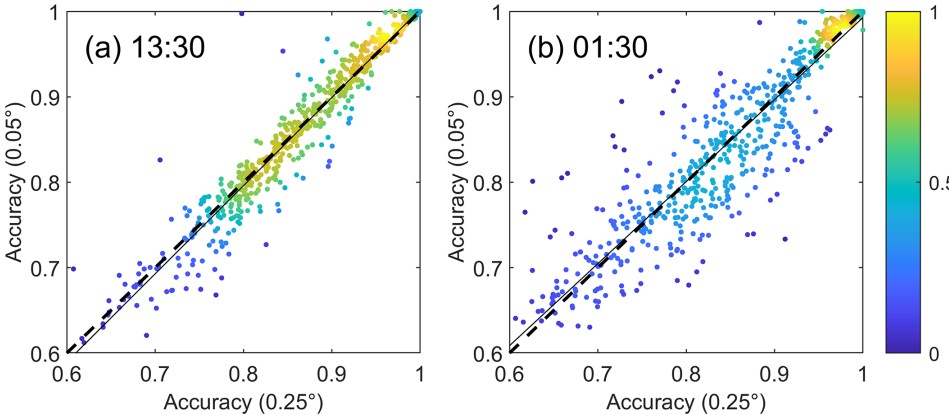


**Figure 6: Scatter plots comparing the downscaled FT product at a 0.05° resolution with the FT product at a 0.25° resolution. The plots show data for (a) ascending orbits and (b) descending orbits. The color bar represents the density of data points.**

In this section, the coarse- and high-resolution FT products were validated against in situ soil temperature data from 44 networks. The scatter plots of accuracy across all ground stations are shown in Fig. 6, with detailed validation results provided

in Table 3. The downscaled products achieved overall accuracies of 87.63% and 83.78% at ascending and descending orbit times, respectively. These accuracies are comparable to those of the original products, which had overall accuracies of 87.72% and 84.08%.



For networks experiencing FT cycles, the accuracies of the high-resolution products are 85.91% and 81.55% at ascending and descending orbit times, respectively. For networks without FT phenomena, the accuracies are 97.43% and 97.97%. The color

gradient in the plots of Fig. 6 reflects data density, with points shifting toward yellow as density increases. Additionally, in 65.91% and 56.92% of networks, accuracy exceeds 90% at both ascending and descending orbit times for both FT products. These findings suggest that the data points are most densely concentrated in the range where accuracy values lie between 0.9 and 1. Both ascending and descending orbit results closely align with the one-to-one line, indicating that the accuracy of the FT classification remains largely unchanged after the downscaling process. Despite the enhanced spatial resolution, the

downscaled product preserves the accuracy and consistency of the original resolution.

These results are consistent with the conventional understanding of downscaling, which aims to enhance spatial resolution without compromising the accuracy of the original product. As a result, the downscaled FT product provides higher spatial resolution and more detailed information while maintaining the integrity and quality of the original observations.

**Table 3 Validation results for the coarse- and high-resolution FT products across various networks. Networks 1 to 27 typically**
**experience FT cycles, while networks 28 to 44 do not exhibit FT phenomena. "Num" represents the number of FT product samples that matched in situ measurements at the corresponding sampling times and were used for validation.**

| No | Network | Ascend | | | Descend | | |
|----|---------|--------|----------|----------|---------|----------|----------|
| | | Num | Accuracy (0.05°) | Accuracy (0.25°) | Num | Accuracy (0.05°) | Accuracy (0.25°) |
| 1 | BNZ-LTER | 6,977 | 90.28% | 91.89% | 7,641 | 89.45% | 95.75% |
| 2 | CTP_SMTMN | 22,654 | 73.59% | 74.42% | 25,433 | 87.04% | 84.45% |
| 3 | FLUXNET-AMERIFLUX | 5,981 | 99.57% | 99.52% | 5,726 | 99.25% | 99.09% |
| 4 | FMI | 5,395 | 66.71% | 75.94% | 5,701 | 69.20% | 75.39% |
| 5 | FR_Aqui | 2,947 | 99.97% | 99.90% | 3,018 | 100.00% | 100.00% |
| 6 | GTK | 1,686 | 83.33% | 82.27% | 1,124 | 84.16% | 88.52% |
| 7 | HOAL | 1,450 | 93.10% | 93.03% | 1,568 | 98.47% | 96.49% |
| 8 | HOBE | 6,931 | 91.43% | 88.57% | 7,916 | 97.02% | 95.89% |
| 9 | HYDROL-NET_PERUGIA | 1,101 | 95.64% | 95.37% | 985 | 95.94% | 95.94% |
| 10 | HiWATER_EHWSN | 43 | 100.00% | 100.00% | 41 | 100.00% | 100.00% |
| 11 | KHOREZM | 25 | 100.00% | 100.00% | 27 | 100.00% | 100.00% |
| 12 | MAQU | 12,050 | 76.67% | 77.35% | 13,019 | 80.71% | 76.34% |
| 13 | MOL-RAO | 4,268 | 94.80% | 94.24% | 5,152 | 95.57% | 95.38% |
| 14 | NGARI | 7,699 | 81.60% | 80.70% | 8,188 | 69.89% | 69.42% |
| 15 | NVE | 304 | 69.41% | 82.57% | 0 | / | / |
| 16 | RISMA | 13,435 | 93.80% | 92.50% | 13,772 | 89.34% | 92.35% |
| 17 | Ru_CFR | 1,182 | 82.66% | 85.87% | 896 | 89.06% | 84.71% |



| | | | | | | | |
|---|---|---|---|---|---|---|---|
| 18 | SMN-SDR | 4,335 | 80.16% | 81.34% | 4,816 | 84.88% | 85.24% |
| 19 | SMOSMANIA | 43,811 | 99.45% | 98.98% | 41,441 | 99.56% | 99.11% |
| 20 | SONTE-China | 662 | 81.27% | 82.78% | 755 | 84.11% | 86.23% |
| 21 | TibetObs-Naqu | 24,814 | 73.81% | 74.77% | 27,829 | 85.90% | 83.39% |
| 22 | TibetObs-Pali | 5,427 | 91.84% | 90.82% | 5,645 | 80.96% | 84.00% |
| 23 | TxSON | 5,879 | 100.00% | 100.00% | 5,476 | 100.00% | 99.95% |
| 24 | USDA-ARS | 11,926 | 96.01% | 96.43% | 12,014 | 94.93% | 95.04% |
| 25 | SCAN | 486,602 | 94.20% | 93.95% | 506,250 | 91.18% | 89.99% |
| 26 | SNOTEL | 1,189,704 | 80.81% | 81.94% | 1,332,948 | 75.23% | 75.85% |
| 27 | USCRN | 235,292 | 93.26% | 92.29% | 238,498 | 89.39% | 89.31% |
| Overall accuracy for networks experiencing FT cycles | | 2,102,580 | 85.91% | 86.41% | 2,275,879 | 81.55% | 81.60% |
| 28 | AACES | 1,674 | 100.00% | 100.00% | 1,721 | 99.88% | 100.00% |
| 29 | ARM | 47,249 | 98.94% | 98.96% | 43,150 | 98.55% | 98.10% |
| 30 | BIEBRZA_S-1 | 631 | 90.49% | 89.22% | 661 | 96.52% | 96.37% |
| 31 | DAHRA | 1,233 | 100.00% | 100.00% | 1,009 | 100.00% | 100.00% |
| 32 | MySMNet | 111 | 100.00% | 100.00% | 103 | 100.00% | 100.00% |
| 33 | OZNET | 17,128 | 100.00% | 99.99% | 16,939 | 99.94% | 100.00% |
| 34 | REMEDHUS | 16,943 | 99.98% | 99.89% | 17,273 | 99.95% | 99.86% |
| 35 | RSMN | 24,231 | 95.80% | 95.21% | 15,957 | 96.37% | 95.57% |
| 36 | SASMAS | 2,094 | 100.00% | 100.00% | 2,069 | 100.00% | 100.00% |
| 37 | SOILSCAPE | 5,475 | 90.47% | 90.48% | 5,029 | 97.77% | 97.63% |
| 38 | SWEX_POLAND | 876 | 93.72% | 94.41% | 891 | 89.23% | 90.68% |
| 39 | TAHMO | 1,583 | 100.00% | 99.68% | 1,111 | 100.00% | 100.00% |
| 40 | TERENO | 5,304 | 92.48% | 89.57% | 6,529 | 97.33% | 96.88% |
| 41 | TWENTE | 13,339 | 95.80% | 93.94% | 15,136 | 99.09% | 98.39% |
| 42 | VAS | 365 | 99.45% | 99.18% | 298 | 100.00% | 100.00% |
| 43 | WSMN | 370 | 25.14% | 1.35% | 456 | 37.72% | 2.63% |
| 44 | XMS-CAT | 6,763 | 95.00% | 94.69% | 6,886 | 90.01% | 85.81% |
| Overall accuracy for networks without FT phenomena | | 145,369 | 97.43% | 96.97% | 135,218 | 97.97% | 97.30% |
| Overall accuracy | | 2,247,949 | 87.63% | 87.72% | 2,411,097 | 83.78% | 84.08% |




### 4.3 FT dynamics and trends

#### 4.3.1 Trend analysis of frost days

After validating the FT dataset's accuracy with in situ observations, we calculated the average number of frost days and their trends over the period 2003–2023. Fig. 7(a) illustrates the spatial distribution of the annual average frost days in the Northern Hemisphere. In high-latitude regions (north of 45°N), the average frost days are approximately 187.8 ± 12.7 days (spatial standard deviation). In contrast, frost days in lower-latitude regions vary due to seasonal shifts.

While frost days generally increase with latitude, this pattern is not globally consistent. For example, the Qinghai-Tibet Plateau, despite its relatively low latitude, experiences a higher number of frost days due to its high elevation. In the Southern
Hemisphere, soil freezing is rare, except in localized regions along the Andes Mountains, where freezing occasionally occurs under specific climatic conditions.

The spatial trends of annual frost days are shown in Figure 7(b). MK trend analysis identifies a decreasing trend in approximately 14.35% of global land areas, with 2.67% exhibiting statistically significant declines. This trend reflects the impact of climate warming, especially across much of the Eurasian continent and Alaska, with particularly obvious effects in
Russia and the Qinghai-Tibet Plateau.

In contrast, about 11.17% of the global land area shows an increasing trend in frost days, of which 1.55% are statistically significant. Regions with increasing frost days include North America and West Asia. These opposing trends underscore the complexity of regional climate dynamics, revealing that while many areas are warming with fewer frost days, localized cooling in some areas results in more frequent freezing events.
The analysis of frost day trends demonstrates the utility of the FT dataset while providing valuable insights into regional variations in climate change impacts.

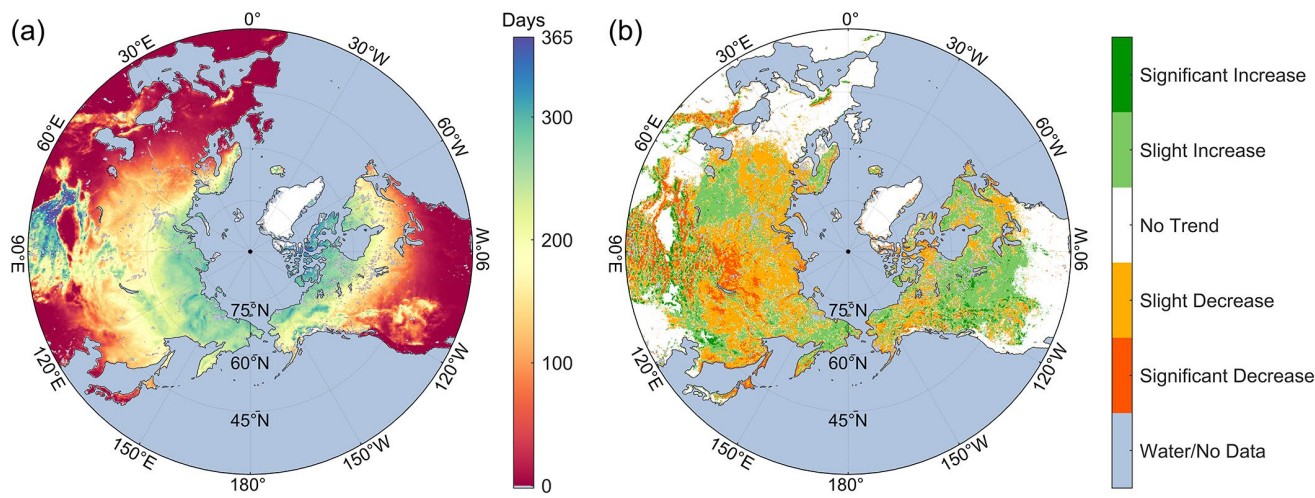





**Figure 7: Northern Hemisphere maps showing (a) average annual frost days and (b) trend analysis results for the 0.05° downscaled FT product during the descending satellite pass from 2003 to 2023.**

### 4.3.2 Trend analysis of freeze onset date

A statistical analysis was performed to examine the average freeze onset date and its trend from 2003 to 2023. Fig. 8(a) shows the spatial distribution of the annual average freeze onset date across the Northern Hemisphere. Soil freezing predominantly occurs in the Northern Hemisphere, where the average freeze onset date in high-latitude regions is approximately $240.3 \pm 7.2$ days (spatial standard deviation). A general trend is observed, with freezing beginning earlier in higher-latitude regions compared to lower-latitude areas. However, this pattern is not uniform. On the Qinghai-Tibet Plateau, the high elevation causes freezing to occur earlier, leading to lower freeze onset dates. In the Southern Hemisphere, soil freezing is limited to small areas, primarily along the Andes Mountains, where freezing also begins earlier in the year.

Fig. 8(b) illustrates the spatial patterns of trends in freeze onset dates. The MK trend analysis identifies a decreasing trend in freeze onset dates approximately 9.10% of the global land area, of which 1.22% exhibits statistically significant reductions. A decrease in freeze onset date indicates an earlier start to freezing in these regions, with the most pronounced trends observed in eastern Russia. Conversely, approximately 7.57% of the global land area demonstrates an increasing trend in freeze onset dates, with 0.91% exhibiting statistically significant increases. An increase in freeze onset date reflects a delayed start to freezing. These regions are more geographically dispersed, reflecting the widespread influence of global warming on soil FT dynamics.

Investigating frost day and freeze onset trends demonstrate the applicability of the FT dataset and provides deeper insight into regional variations in climate change effects on global FT patterns.

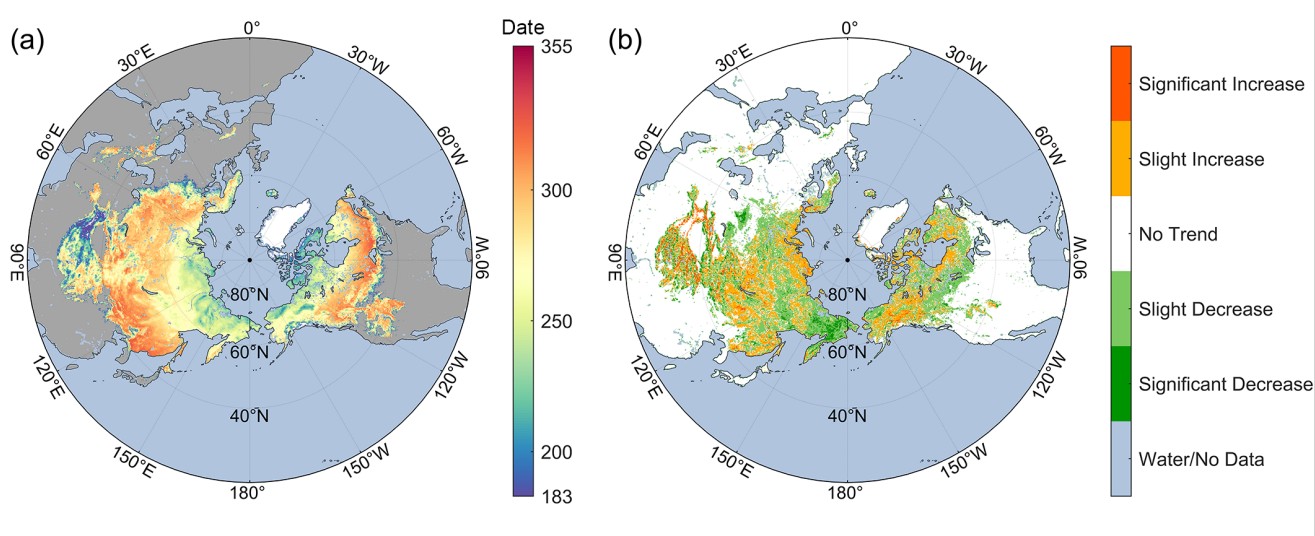

**Figure 8: Northern Hemisphere maps showing (a) annual average freeze onset date and (b) trend analysis results for the 0.05° downscaled FT product from 2003 to 2023.**



## 4.4 Discussion

### 4.4.1 Enhanced capabilities of high-resolution FT products

This study utilizes passive microwave and optical observations to achieve high-resolution detection of FT states. Initially, passive microwave sensors were used to monitor global surface FT states, generating an original FT dataset at a 0.25° resolution. Due to their minimal susceptibility to atmospheric conditions, such as cloud cover and aerosols, these sensors enable continuous tracking of FT transitions across diverse landscapes and climatic zones globally. Furthermore, the associated discrimination algorithm demonstrates high classification accuracy, rendering it suitable for global-scale applications. However, the original dataset's coarse spatial resolution of 0.25°, combined with inherent seams in microwave sensor data, limits its capacity to capture detailed surface variations. In contrast, optical observations, including LST and albedo, offer higher spatial resolution and provide more detailed surface information. Additionally, the development of seamless, long-term data products not only addresses the data seams in the original dataset but also maintains a short revisit interval, enabling daily surface monitoring.

Therefore, to integrate the advantages of both data sources, a bivariate regression model was developed, effectively downscaling the original 0.25° resolution dataset to a finer 0.05° resolution. This improved resolution allows for a more detailed representation of FT states, facilitating analyses of regional FT dynamics, particularly in areas with complex terrain, diverse vegetation, or significant seasonal variations. As shown in Fig. 9, the 0.05° resolution FT product reveals finer details of FT cycles that were previously undetectable in the 0.25° dataset. These enhancements are critical for research requiring precise identification of localized surface condition changes, such as those affecting soil moisture and ecosystem carbon dynamics.

The methodology employed in this study not only eliminates seams in the original FT dataset but also integrates data from multiple sources, including MODIS optical products and microwave TB observations. This fusion provides a robust and continuous long-term FT record, essential for understanding global FT cycles and their environmental implications. Moreover, the enhanced spatial resolution enables detailed analyses of changes in local FT conditions, offering valuable insights for hydrological modeling, climate research, and ecosystem management. These advancements expand the applicability of the FT record, contributing significantly to diverse scientific fields.



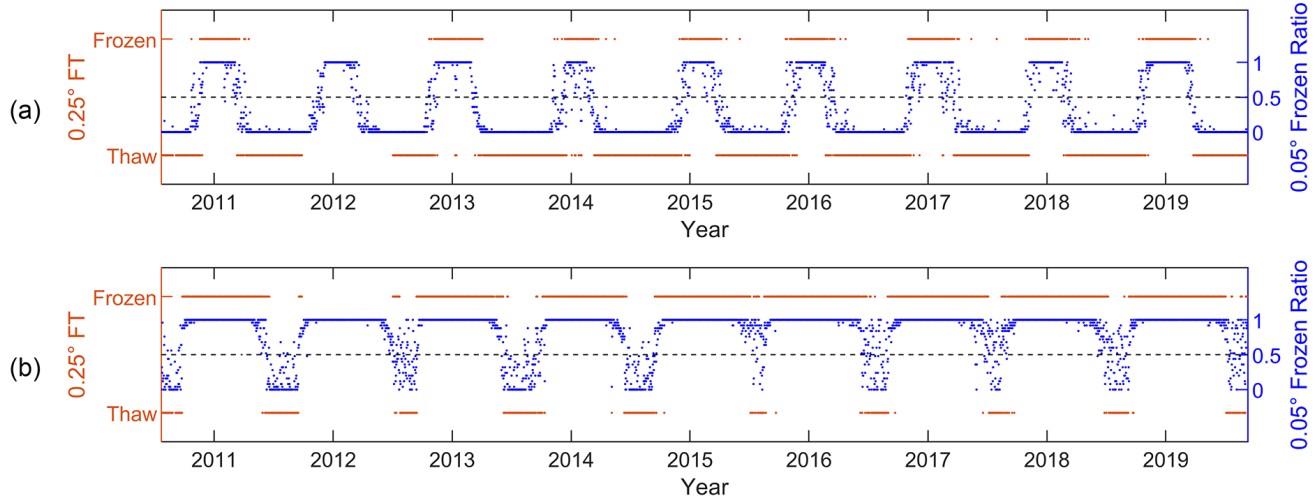


**Figure 9: Time series plots comparing the 0.25° FT product values and the 0.05° frozen ratios in the downscaled product at the SQ21 station of the NGARI network, distinguished by (a) daytime and (b) nighttime observations.**

### 4.4.2 Uncertainties associated with ground validation

Ground temperature data serve as a reliable basis for validating satellite-derived FT products. In this study, soil temperature

measurements at 0–5 cm depth were used for validation rather than surface-level (0 cm) measurements due to their superior representation of near-surface soil FT states. While 0°C is commonly employed as the threshold for distinguishing frozen and thawed states in validation using in situ temperature data, this criterion may not always accurately reflect actual ground conditions. Various factors, including wind, vegetation cover, and snow cover, can influence the true ground state even when surface temperatures are at or below 0°C.

Currently, there is no universally accepted standard for defining absolute temperature boundaries for frozen and thawed ground. However, soil temperature measurements at a depth of 0–5 cm provide a more reliable indicator than alternatives, such as air temperature or measurements from deeper soil layers, as they more directly capture the near-surface state. Despite this, under certain conditions, such as areas with strong winds or heavy snow cover, this approach may lead to classification errors, particularly when the ground thawing process is disrupted at temperatures near 0°C. To address these limitations, future studies

should concentrate on identifying optimal soil depths and establishing more precise temperature thresholds to enhance validation accuracy.

### 4.4.3 Limitations and challenges

Although this study has effectively generated high-resolution FT data by integrating passive microwave observations with optical datasets, further refinements are required to address the remaining challenges. First, despite the resolution enhancement

to 0.05°, it is still relatively coarse for applications demanding finer-scale detail. In complex surface environments, this





resolution may not adequately capture localized variations in FT states. To reduce the influence of spatial heterogeneity, higher-resolution optical observations are necessary. For instance, utilizing a 1-kilometer LST and albedo dataset for downscaling would align the resulting FT record with this finer spatial resolution. While 1-kilometer LST datasets are available, the development of more stable and higher-resolution LST products remains critical. The quality of the downscaled FT dataset is

closely reliant on the quality of the optical observations employed. Future research should prioritize the use of seamless, high-quality optical datasets with short revisit intervals and higher spatial resolution to produce more precise and reliable FT classification products.

Second, the accuracy of the downscaled FT dataset is inherently determined by FT discrimination algorithms, which rely on lower-resolution microwave remote sensing. As outlined in Sect. 4.2, the downscaling process preserves the accuracy

characteristics of the original FT data. Therefore, improvements in the FT discrimination algorithms are necessary for achieving further gains in classification accuracy.

Additionally, during the data fusion process, combining optical observations with microwave data, various factors may affect the correlation coefficients and the performance of the linear regression model. These factors include land cover types, vegetation conditions, terrain elevation, and seasonal variations, all of which can introduce instability in model performance

and impact FT classification accuracy. To address these limitations, future studies should incorporate additional surface parameters, including the Normalized Difference Vegetation Index (NDVI), to improve the model's adaptability and accuracy across diverse surface conditions.

## 5 Data availability

The global 0.05° near-surface soil FT state dataset (2002–2023) is freely available at the National Tibetan Plateau / Third Pole

Environment Data Center. The dataset can be accessed via https://doi.org/10.11888/Cryos.tpdc.301551 or https://cstr.cn/18406.11.Cryos.tpdc.301551 (Zhao et al., 2024b).

## 6 Conclusion

This study developed a comprehensive, high-resolution global dataset of surface FT states by integrating multi-source remote sensing data. The primary objective was to enhance spatial resolution and provide more detailed global monitoring of FT states.

Coarse-resolution FT data were initially derived from passive microwave TBs, while long-term ATI data were calculated using optical observations. By leveraging the complementary strengths of passive microwave and optical remote sensing products, a high-resolution daily near-surface FT dataset was produced, offering a finer representation of surface FT conditions.

Subsequently, the coarse- and high-resolution FT products were validated using ground-based in situ observations. This validation facilitated a thorough evaluation of accuracy changes associated with the downscaling process and enabled a

comprehensive trend analysis on a global scale using the enhanced-resolution FT records.

The findings have significant implications for understanding and monitoring ecological and hydrological responses to climate dynamics. The analysis revealed intricate patterns of frost days and freeze onset dates, demonstrating the varying regional influences of climate change. The downscaled product, with its enhanced spatial resolution, provided detailed insights into FT dynamics, which are crucial for studies requiring precise identification of local surface condition changes, such as those

impacting soil moisture or ecosystem carbon dynamics.

This integration of multi-source remote sensing data marks a significant advancement in monitoring Earth's surface processes. It demonstrates the potential to improve climate models and other environmental assessments. Furthermore, the proposed methodological framework is adaptable for similar applications globally, enhancing the predictive capabilities for climate-related phenomena and supporting environmental decision-making. By providing a robust and continuous long-term FT record,

this study contributes to the development of hydrological models, climate studies, and ecosystem management, thereby expanding the applicability of the record in diverse scientific fields.

**Author contributions**

TZ and JS conceptualized the research; JS managed project administration, supervised the project, acquired funding, and collected resources; DF and TZ designed the methodology, validated the experimental designs, and wrote the codes; DF, JYZ,

YB, XK, and PY visualized research outcomes and performed formal analysis; DF, JYZ, YB, XK, LJ, ZZ, PY, JBZ and JP curated data; TZ, YR, LJ, ZZ, JBZ, JP, PY, and YL conducted investigations; DF and TZ drafted the manuscript; YR and YL provided guidance and revised the manuscript; All authors participated in discussions and provided guidance and advice throughout the experimental design and data validation process, and all reviewed the manuscript.

**Competing interests**

The contact author has declared that none of the authors has any competing interests.

**Acknowledgments**

We are grateful for the free access to AMSR-E and AMSR2 TB data provided by the National Snow and Ice Data Center (NSIDC, https://www.nsidc.org) and JAXA (https://gportal.jaxa.jp). We also acknowledge the GLASS albedo products from the National Earth System Science Data Center, National Science & Technology Infrastructure of China

(http://www.geodata.cn), MODIS land cover products from the Land Processes Distributed Active Archive Center (LP DAAC, https://lpdaac.usgs.gov/), global in situ soil temperature data from ISMN (https://ismn.earth/), and the global spatiotemporally continuous LST datasets and Tibet-Obs surface soil temperature datasets from the National Tibetan Plateau / Third Pole Environment Data Center (http://data.tpdc.ac.cn).





**Financial support**

This study is funded by the National Key Research and Development Program of China (No. 2021YFB3900104) and the
Fengyun Application Pioneering Project (FY-APP-2022.0305).

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
