# Peer review of "A high-resolution (0.05°) global seamless continuity record (2002–2023) of near-surface soil freeze-thaw states via passive microwave and optical satellite data"

_Earth System Science Data, 2025_

## Author Comment (AC1)

**Response to RC1**

The article is well-structured, and the technique and results are thoroughly explained. However, there are a few important comments:

**Reply:**

Thank you for your positive assessment and thoughtful comments. Our responses to each point are provided below.

**1.** At what depths was in-situ data used to confirm soil freezing beyond 5 cm? Freezing to a depth of 5 cm is a significant criterion for some agricultural activities, but it is insufficient for the majority of hydrological and ecological tasks, particularly in the context of climate change.

**Reply:**

Thank you for your inquiry regarding the depth of in situ measurements used for validating soil freezing. In validating both the coarse-resolution and high-resolution freeze-thaw (FT) datasets, we exclusively relied on ground-based soil temperature data from depths shallower than 5 cm. The initial coarse-resolution FT discrimination is derived from AMSR-E/2 passive-microwave brightness temperatures (TBs) at K-band (18.7 GHz) and Ka-band (36.5 GHz), whose typical penetration depths are approximately 3–5 cm and 0–2 cm, respectively, both under 5 cm. Given these penetration-depth constraints of passive microwave sensing, we confined our validation to shallow (<5 cm) observations to ensure that the in situ data and satellite-derived classifications sample the same sensitive soil layer. We also acknowledge the importance of deeper FT dynamics for climate and ecosystem studies and may investigate remote-sensing approaches for monitoring and analyzing greater freeze depths in future work.

We have revised Sections 2.5 and 4.4.2 to clearly state the depths of ground-based soil temperature measurements used for validation and to explain the rationale for focusing on shallow (<5 cm) layers.

**Changes in manuscript:**

Lines 177-178 in Section 2.5:

"This study selected long-term in situ soil temperature data from 1,027 stations within 44 global networks, all measured at a depth of 0–5 cm to match the penetration depth of the passive microwave observations."

Lines 518-520, 525-527 in Section 4.4.2:

"Additionally, the utilization of 0–5 cm depth data ensures comparability with satellite-based FT

discrimination, since the typical penetration depths of passive microwave observations at 18.7 GHz and 36.5 GHz are approximately 3–5 cm and 0–2 cm, respectively, both under 5 cm."

"However, the use of 0–5 cm soil temperature remains the most physically meaningful for passive microwave FT validation, as it provides a more direct and reliable indicator of the near-surface state than either air temperature or deeper soil temperature measurements."

**2.** How do the authors incorporate data on permafrost distribution into the FT product, both in extended areas and mountainous regions? How do authors account for geographical and temporal variations in permafrost indicators between 2002 and 2023?

**Reply:**

Thank you for your question. We address it in two parts:

First, **we did not incorporate any existing permafrost-distribution maps into our FT dataset.** As shown in Fig. 1, we only used three IGBP land-cover classes as masking layers: water bodies, urban and built-up lands, and snow and ice. No permanent-permafrost map was employed, either in the extended areas or in mountainous regions. This explanation has been added to Section 2.4 of the revised manuscript.

Second, although our 0.05° FT dataset focuses on near-surface soil phase transitions, it enables the derivation of annual FT metrics (e.g. the number of frost days, the number of FT cycles) for each grid cell. Mapping these metrics across diverse landscapes provides a remote sensing approach to assess spatial and temporal variations in permafrost extent from 2002 to 2023.

For instance, **while our dataset does not use any existing permafrost distribution data as input,** the results show spatial agreement between the annual number of frost days derived from our remote sensing approach in 2017 and independently derived permafrost maps over the Qinghai–Tibetan Plateau (Zhao, 2017), as illustrated in Fig. 8 of the revised manuscript. Specifically, the average annual frost days within permafrost-classified pixels is approximately 278.85, indicating a potential spatial correspondence between these two metrics. This qualitative comparison demonstrates that **regions with high annual numbers of frost days, as detected by remote sensing, are generally consistent with areas classified as permafrost in existing reference datasets**, supporting the potential of the annual number of frost days to serve as a valuable proxy for assessing permafrost extent and its spatial variability. This discussion has been added at the end of Section 4.3.1 of the revised manuscript.

**Changes in manuscript:**

Lines 165-168 in Section 2.4:

"The land cover dataset, illustrated in Fig. 1, was specifically utilized to mask out pixels of three IGBP land-cover classes: water bodies, urban and built-up lands, and snow and ice. Furthermore, the corresponding type percentage dataset was used to filter out pixels dominated by large water bodies, which were then explicitly marked in the FT data record. Notably, no permafrost distribution data were used as input or constraints in generating the FT dataset."

Lines 457–465, Figure 8 in Section 4.3.1:

"To further explore the climatic and geocryological significance of this metric, we compared the spatial distribution of annual frost days in 2017 with independently derived permafrost maps over the Qinghai–Tibetan Plateau (Zhao, 2017; Zou et al., 2017), as illustrated in Fig. 8. This qualitative comparison reveals a notable spatial agreement, demonstrating that regions with high annual numbers of frost days, as detected by remote sensing, are generally consistent with areas classified as permafrost in existing reference datasets. Specifically, the average annual frost days within permafrost-classified pixels is approximately 278.85, indicating a potential spatial correspondence between these two metrics. This finding highlights the potential of the annual number of frost days as a valuable proxy for assessing permafrost extent and its spatial variability."

[Figure]

**Figure 8: Maps of (a) permafrost distribution map (Zhao, 2017), and (b) annual frost days derived from the downscaled descending FT product in 2017 over the Qinghai–Tibet Plateau.**

Lines 799–800, 820-822 in Reference:

Zhao, L.: A new map of permafrost distribution on the Tibetan Plateau (2017), National Tibetan Plateau /

Third Pole Environment Data Center [dataset], https://doi.org/10.11888/Geocry.tpdc.270468, 2017.

Zou, D., Zhao, L., Sheng, Y., Chen, J., Hu, G., Wu, T., Wu, J., Xie, C., Wu, X., Pang, Q., Wang, W., Du, E., Li, W., Liu, G., Li, J., Qin, Y., Qiao, Y., Wang, Z., Shi, J., and Cheng, G.: A new map of permafrost distribution on the tibetan plateau, The Cryosphere, 11, 2527–2542, https://doi.org/10.5194/tc-11-2527-2017, 2017.

**3** The authors do not provide a convincing number of point verifications of the FT product in various physiographic conditions using statistical techniques. In addition, a comparison of soil freezing depth inaccuracies in various natural zones is required.

**Reply:**

Thank you for your valuable suggestion.

To address the issue of insufficient point verifications across different physiographic conditions and land cover types, we have added a statistical accuracy table based on the ESA CCI Land Cover 2010 classifications provided by ISMN sites (see Table 4 of the revised manuscript and Response Fig. 1). This table presents classification accuracies of both coarse-resolution (0.25°) and high-resolution (0.05°) products across different land cover types, with separate statistics for ascending and descending satellite passes, thereby reflecting the accuracy under various physiographic conditions and land cover types. This content has been included in Section 4.2 of the revised manuscript.

As for the comparison of soil freezing depth inaccuracies across natural zones, we are initiating a follow-up study that uses the annual total frozen-days metric from our FT dataset to simulate soil freezing depths and assess their variability across different natural regions. A detailed comparison will be presented in future work.

[Figure]

**Response Figure 1:** Line plots of classification accuracy for the 0.25° and 0.05° FT products across different land cover types: (a) ascending passes; (b) descending passes. The x-axis abbreviations are explained in Response Table 1

**Response Table 1:** ESA CCI land cover classification framework.

| | ESA CCI Land Cover Classes | | | ESA CCI Land Cover Classes | |
|---|---|---|---|---|---|
| **Code** | Abbreviated name | Full name | **Code** | Abbreviated name | Full name |
| 10 | CR | Cropland, rainfed | 100 | MTS/HC | Mosaic tree and shrub (>50%) / herbaceous cover (<50%) |
| 11 | CR/HC | Cropland, rainfed / Herbaceous cover | 110 | MHC/TS | Mosaic herbaceous cover (>50%) / tree and shrub (<50%) |
| 12 | CR/TS | Cropland, rainfed / Tree or shrub cover | 120 | SH | Shrubland |
| 20 | CI | Cropland, irrigated or post-flooding | 130 | GR | Grassland |
| 30 | MC/NV | Mosaic cropland (>50%) / natural vegetation (tree, shrub, herbaceous cover) (<50%) | 140 | LM | Lichens and mosses |
| 40 | MNV/CR | Mosaic natural vegetation (tree, shrub, herbaceous cover) (>50%) / cropland (<50%) | 150 | SV | Sparse vegetation (tree, shrub, herbaceous cover) (<15%) |
| 50 | TBE | Tree cover, broadleaved, evergreen, Closed to open (>15%) | 180 | SHF | Shrub or herbaceous cover, flooded, fresh/saline/brakish water |
| 60 | TBD | Tree cover, broadleaved, deciduous, | 190 | UA | Urban areas |

| No | | Description | No | | Description |
|---|---|---|---|---|---|
| | | Closed to open (>15%) | | | |
| 61 | TBD-C | Tree cover, broadleaved, deciduous, Closed (>40%) | 200 | BA | Bare areas |
| 62 | TBD-O | Tree cover, broadleaved, deciduous, Open (15-40%) | 201 | CBA | Consolidated bare areas |
| 70 | TNE | Tree cover, needleleaved, evergreen, closed to open (>15%) | 210 | WT | Water |
| 90 | TMX | Tree cover, mixed leaf type (broadleaved and needleleaved) | 220 | PSI | Permanent snow and ice |

**Changes in manuscript:**

Lines 412-415 in Section 4.2:

In addition, Table 4 summarizes the classification accuracies of both products across various land cover types for both ascending and descending passes. The land cover types are defined by the ESA CCI Land Cover 2010 classification values provided by the ISMN sites, enabling a more comprehensive assessment of performance under different physiographic conditions.

Lines 430-432, Table 4 in Section 4.2:

**Table 4 Validation results for the coarse- and high-resolution FT products across various land cover types. Land cover types are sourced from the ESA CCI Land Cover 2010 classification values provided within the ISMN site dataset.**

| No | Land cover type | Ascend | | | Descend | | |
|---|---|---|---|---|---|---|---|
| | | Num | Accuracy (0.05°) | Accuracy (0.25°) | Num | Accuracy (0.05°) | Accuracy (0.25°) |
| 1 | Cropland, rainfed | 275,743 | 94.66% | 94.37% | 277,149 | 92.71% | 92.26% |
| 2 | Cropland, rainfed / Herbaceous cover | 10,129 | 90.79% | 80.11% | 10,157 | 83.01% | 62.56% |
| 3 | Cropland, rainfed / Tree or shrub cover | 12,127 | 99.54% | 99.20% | 11,336 | 99.07% | 98.42% |
| 4 | Cropland, irrigated or post-flooding | 5,162 | 99.99% | 99.91% | 5,217 | 99.93% | 99.81% |
| 5 | Mosaic cropland (>50%) / natural vegetation (tree, shrub, herbaceous cover) (<50%) | 4,896 | 95.70% | 96.12% | 5,439 | 93.14% | 93.47% |
| 6 | Mosaic natural vegetation (tree, shrub, herbaceous cover) (>50%) / cropland (<50%) | 25,702 | 91.44% | 89.54% | 27,613 | 82.67% | 79.58% |
| 7 | Tree cover, broadleaved, | 7,466 | 99.14% | 94.58% | 12,210 | 79.45% | 64.53% |

| | | | | | | | |
|---|---|---|---|---|---|---|---|
| | evergreen, Closed to open (>15%) | | | | | | |
| 8 | Tree cover, broadleaved, deciduous, Closed to open (>15%) | 12,658 | 94.14% | 93.40% | 12,094 | 94.76% | 93.53% |
| 9 | Tree cover, broadleaved, deciduous, Closed (>40%) | 15,445 | 90.60% | 91.34% | 14,557 | 91.21% | 90.13% |
| 10 | Tree cover, broadleaved, deciduous, Open (15-40%) | 198 | 100.00% | 100.00% | 198 | 100.00% | 100.00% |
| 11 | Tree cover, needleleaved, evergreen, closed to open (>15%) | 898,729 | 81.77% | 82.81% | 996,959 | 76.49% | 76.84% |
| 12 | Tree cover, mixed leaf type (broadleaved and needleleaved) | 10,529 | 83.71% | 83.61% | 12,351 | 79.36% | 84.00% |
| 13 | Mosaic tree and shrub (>50%) / herbaceous cover (<50%) | 22,384 | 90.10% | 90.79% | 24,002 | 85.67% | 90.28% |
| 14 | Mosaic herbaceous cover (>50%) / tree and shrub (<50%) | 5,159 | 90.48% | 91.17% | 4,955 | 92.15% | 93.02% |
| 15 | Shrubland | 217,565 | 89.63% | 89.83% | 236,376 | 85.49% | 85.47% |
| 16 | Grassland | 596,806 | 87.98% | 88.41% | 626,309 | 84.49% | 84.47% |
| 17 | Lichens and mosses | 2,711 | 92.62% | 92.88% | 3,153 | 86.24% | 93.78% |
| 18 | Sparse vegetation (tree, shrub, herbaceous cover) (<15%) | 25,240 | 88.56% | 88.26% | 28,065 | 83.54% | 87.57% |
| 19 | Shrub or herbaceous cover, flooded, fresh/saline/brakish water | 16,913 | 83.87% | 85.76% | 19,753 | 80.89% | 85.64% |
| 20 | Urban areas | 41,406 | 92.64% | 92.07% | 37,230 | 89.29% | 89.39% |
| 21 | Bare areas | 2,955 | 93.10% | 93.49% | 3,178 | 89.26% | 90.64% |
| 22 | Consolidated bare areas | 642 | 85.41% | 85.31% | 665 | 67.48% | 65.08% |
| 23 | Water | 5,213 | 81.31% | 81.96% | 6,120 | 73.57% | 74.23% |
| 24 | Permanent snow and ice | 0 | / | / | 0 | / | / |

**4.** Can you explain the lack of freeze-thaw data at 0.25° resolution in Fig. 4?

**Reply:**

Thank you for this question.

The lack of 0.25° freeze-thaw (FT) data shown in Fig. 4(a) and (b) arises from the satellite's inherent

orbital swath gaps in daily AMSR-E/2 passive-microwave brightness-temperature (TB) observations. Because each overpass only covers a limited swath, grid cells outside that swath on any given day lack valid TB inputs, making FT discrimination at 0.25° impossible and resulting in missing values. The approximately 2-day revisit interval of AMSR-E/2 causes the location of valid observations to shift from one day to the next. Although this results in missing measurements for certain pixels on some dates, each pixel still accumulates sufficient valid observations over the course of a year.

In contrast, Fig. 4(c) and (d) show our 0.05° high-resolution downscaled product, which contains no data gaps because our pixel-by-pixel linear-regression downscaling automatically skips any dates with missing inputs. Specifically, for each 0.25° pixel, regression fitting was performed using only those dates with valid 0.25° FT data, together with the corresponding 0.05° optical data on those dates. The 0.05° optical data are spatiotemporally continuous, so regression fitting is always possible on any date with valid 0.25° FT observations. For example, if a given pixel has valid 0.25° FT data on only 200 out of 365 days, the regression coefficients are derived solely from those 200 days. We then apply the fitted model to the full year of 0.05° optical data, thereby generating a seamless 0.05° FT dataset.

A detailed description of this linear-regression downscaling approach and how it addresses missing data is provided in Section 3.3 of the manuscript. Additionally, Section 4.1 provides a brief description of the data gaps present in the 0.25° product, while highlighting that the 0.05° downscaled product contains no such gaps.

**Changes in manuscript:**

Lines 259, 267, 270-272 in Section 3.3:

"where the coefficients a, b and c are determined through linear regression fitting."

"Linear regression fitting was performed on these three data vectors of each pixel, resulting in six coefficient matrices for $a$, $b$, and $c$ at the ascending and descending times:"

"where the abbreviations "HR" and "LR" refer to high and low resolution, respectively. Regression fitting was performed using only those dates with valid 0.25° FT data, together with the corresponding 0.05° optical data on those dates."

Lines 361-363, 370-378 in Section 4.1:

"The left panel of Fig. 4 presents a comparison between the original 0.25° FT discrimination product and the downscaled FT discrimination product for January 1, 2019, at 01:30 (descending orbit)."

"In addition to the classification of surface conditions, another important feature of the dataset is its treatment of missing data. In the original 0.25° FT product, data gaps occur due to the inherent swath gaps

of the AMSR-E/2 satellites, resulting in missing values for some grid cells on certain days. By contrast, our downscaling approach for the 0.05° product effectively overcomes this limitation. For each 0.25° pixel, regression fitting is performed using only those dates with valid 0.25° FT data, together with the corresponding 0.05° optical data on those dates. As the 0.05° optical data are spatiotemporally continuous, regression can always be performed on any date with valid 0.25° FT observations. The fitted regression model is then applied to the complete year of 0.05° optical data, resulting in a seamless, gap-free 0.05° FT dataset. This missing-data treatment in the downscaling process ensures that the high-resolution FT product contains no data gaps, as demonstrated in Fig. 4."

---

## Author Comment (AC2)

**Response to RC2**

This study aims to improve the spatial resolution of FT detection products without sacrificing accuracy, using passive microwave-based FT data from AMSR-E/2 TB through the DFA. Downscaling indicators, such as MODIS-based LST and ATI, are used to integrate soil moisture information. The downscaled FT records were validated with in-situ soil temperatures, assessing accuracy changes post-downscaling. Trend analyses of the high-resolution FT records capture detailed dynamics, meeting the growing spatial and temporal resolution needs of GCOS for FT monitoring. This study provides valuable FT data for enhancing cryospheric and ecological research, offering an important resource for future studies. The study is well written and the methods used are rigorous. I recommend a minor revision to address some of my comments.

**Reply:**

Thank you for your thorough and encouraging evaluation of our work. In line with your recommendation for a minor revision, we have carefully reviewed the manuscript and implemented the following improvements:

**#1** In the study it categorizes up to 17 land cover types, each representing a distinct vegetation type. However, I believe relying solely on satellite images to build FT products may oversimplify the process. It would be beneficial to incorporate the biogeochemical mechanisms specific to each vegetation type (e.g., forest, savanna, grassland) when determining FT. Could you provide further clarification on how these mechanisms are integrated into the analysis?

**Reply:**

Thank you very much for your valuable comments.

We fully agree with your point that freeze-thaw (FT) processes differ significantly among various vegetation types due to their distinct biogeochemical mechanisms. The near-surface soil FT process involves dynamic changes in soil and vegetation moisture states, which lead to significant variations in soil dielectric properties. Specifically, when surface temperatures drop below freezing, the liquid water in the soil gradually freezes into ice; conversely, as temperatures rise above freezing, the ice melts back into liquid water. Due to the substantial difference in dielectric constants between liquid water and ice, this moisture state transition causes pronounced changes in dielectric properties, a phenomenon that is particularly complex across different vegetation types.

Passive microwave remote sensing technology effectively captures these changes in soil dielectric

properties. Therefore, this study introduces the quasi-emissivity (Qe) index in FT discrimination based on passive microwave observations, defined as the ratio of 18.7 GHz horizontally polarized TB ($TB_{18.7H}$) to 36.5 GHz vertically polarized TB ($TB_{36.5V}$). Qe comprehensively characterizes the dielectric property changes occurring throughout the FT process, reflecting dynamic changes in soil and vegetation moisture states. Consequently, this indicator can indirectly capture the biophysical status of vegetation and soil conditions, thus considering the distinct biogeochemical mechanisms specific to different vegetation types during the FT process.

While our current approach does not explicitly quantify these mechanisms for each vegetation type, the satellite-derived Qe index integrates these effects by capturing the combined changes in soil and vegetation dielectric properties. We appreciate your suggestion and may consider incorporating vegetation-specific parameters and biogeochemical data in future studies to better characterize the distinct mechanisms of different vegetation types during the FT process.

We have expanded the description of our approach using quasi-emissivity to capture dynamic changes in soil and vegetation moisture states. This clarification has been added to Section 3.1 of the revised manuscript.

**Changes in manuscript:**

Lines 202–216 in Section 3.1:

"The DFA is a surface FT discrimination method developed using AMSR-E and AMSR2 data, demonstrating high accuracy compared to existing FT products. Near-surface FT variations are closely associated with soil temperature and moisture, which are reflected by the TB at 36.5 GHz in vertical polarization ($TB_{36.5V}$) and the Quasi-emissivity (Qe).

Qe is defined as the ratio of the TB at 18.7 GHz in horizontal polarization ($TB_{18.7H}$) to $TB_{36.5V}$. This ratio serves as an indicator of soil moisture, as microwave emission at these frequencies is highly sensitive to changes in water content. The near-surface soil FT process involves dynamic changes in soil and vegetation moisture states, which lead to significant variations in soil dielectric properties. Due to the substantial difference in dielectric constants between liquid water and ice, these phase transitions cause pronounced dielectric changes, a phenomenon further complicated by the distinct biogeochemical mechanisms of different vegetation types. The Qe index introduced in this study comprehensively characterizes the combined dielectric effects of both soil and vegetation during the FT process. Thus, Qe not only reflects soil moisture variations, but also indirectly captures the biophysical status of vegetation and soil conditions, enabling consideration of the unique biogeochemical processes associated with various vegetation types during the FT process.

Therefore, $TB_{36.5V}$ and Qe were selected as key parameters for FT discrimination (Zhao et al., 2011).

The DFA is further parameterized to separately detect FT status during ascending and descending orbits (Wang et al., 2019b), as expressed by the following equations:"

**#2** While the validation provides an overall accuracy metric, it may not be sufficient to establish robustness. I suggest conducting validation separately within different landscapes where FT characteristics vary significantly. Additionally, comparing results against site-level observations would strengthen the validation process. Including more statistical measures would further enhance the robustness of the findings.

**Reply:**

Thank you for your valuable comments.

Regarding the overall accuracy metric and other statistical measures, since our FT product is a binary classification (frozen/thawed) rather than continuous decimal values, traditional regression metrics such as $R^2$ and RMSE cannot be applied. Therefore, we adopted classification accuracy as the primary metric for validation of our downscaled FT dataset.

Regarding comparing results against site-level observations, we conducted comprehensive validation against in situ soil temperature data from 44 global ground observation networks, covering over 1,000 stations. We carefully selected ground measurements temporally matched to satellite observations to compare the classified frozen/thawed states. The results demonstrate that the downscaled products maintain accuracy comparable to the original coarse-resolution products, with overall accuracies exceeding 83% across multiple networks. Detailed statistics and scatter plots are presented in Section 3.4 (Figure 6 and Table 3) of the manuscript.

In response to the suggestion of conducting validation separately within different landscapes, we have added a statistical accuracy table grouped by land cover types derived from the ESA CCI Land Cover 2010 dataset provided by ISMN sites (see Table 4 of the revised manuscript and Response Fig. 1). This table details the classification accuracies of both 0.25° and 0.05° resolution datasets across different land cover types, effectively capturing product performance variations under diverse physiographic conditions. This content will be incorporated into Section 4.2 to further enhance the robustness of the validation.

We sincerely appreciate your constructive comments, which have helped strengthen the scientific rigor of our study.

[Figure]

**Response Figure 1.** Line plots of classification accuracy for the 0.25° and 0.05° FT products across different land cover types: (a) ascending passes; (b) descending passes. The x-axis abbreviations are explained in Response Table 1.

**Response Table 1** ESA CCI land cover classification framework.

| ESA CCI Land Cover Classes | | | ESA CCI Land Cover Classes | | |
|---|---|---|---|---|---|
| **Code** | Abbreviated name | Full name | **Code** | Abbreviated name | Full name |
| 10 | CR | Cropland, rainfed | 100 | MTS/HC | Mosaic tree and shrub (>50%) / herbaceous cover (<50%) |
| 11 | CR/HC | Cropland, rainfed / Herbaceous cover | 110 | MHC/TS | Mosaic herbaceous cover (>50%) / tree and shrub (<50%) |
| 12 | CR/TS | Cropland, rainfed / Tree or shrub cover | 120 | SH | Shrubland |
| 20 | CI | Cropland, irrigated or post-flooding | 130 | GR | Grassland |
| 30 | MC/NV | Mosaic cropland (>50%) / natural vegetation (tree, shrub, herbaceous cover) (<50%) | 140 | LM | Lichens and mosses |
| 40 | MNV/CR | Mosaic natural vegetation (tree, shrub, herbaceous cover) (>50%) / cropland (<50%) | 150 | SV | Sparse vegetation (tree, shrub, herbaceous cover) (<15%) |
| 50 | TBE | Tree cover, broadleaved, evergreen, Closed to open (>15%) | 180 | SHF | Shrub or herbaceous cover, flooded, fresh/saline/brakish water |
| 60 | TBD | Tree cover, broadleaved, deciduous, | 190 | UA | Urban areas |

| No | | | |
|----|------|-------------------------------------------------------------------|-------------------|
| 61 | TBD-C | Closed to open (>15%) Tree cover, broadleaved, deciduous, Closed (>40%) | 200 BA — Bare areas |
| 62 | TBD-O | Tree cover, broadleaved, deciduous, Open (15-40%) | 201 CBA — Consolidated bare areas |
| 70 | TNE | Tree cover, needleleaved, evergreen, closed to open (>15%) | 210 WT — Water |
| 90 | TMX | Tree cover, mixed leaf type (broadleaved and needleleaved) | 220 PSI — Permanent snow and ice |

**Changes in manuscript:**

Lines 412-415 in Section 4.2:

In addition, Table 4 summarizes the classification accuracies of both products across various land cover types for both ascending and descending passes. The land cover types are defined by the ESA CCI Land Cover 2010 classification values provided by the ISMN sites, enabling a more comprehensive assessment of performance under different physiographic conditions.

Lines 430-432, Table 4 in Section 4.2:

**Table 4 Validation results for the coarse- and high-resolution FT products across various land cover types. Land cover types are sourced from the ESA CCI Land Cover 2010 classification values provided within the ISMN site dataset.**

| No | Land cover type | Ascend | | | Descend | | |
|----|-----------------|--------|-----------------|-----------------|---------|-----------------|-----------------|
| | | Num | Accuracy (0.05°) | Accuracy (0.25°) | Num | Accuracy (0.05°) | Accuracy (0.25°) |
| 1 | Cropland, rainfed | 275,743 | 94.66% | 94.37% | 277,149 | 92.71% | 92.26% |
| 2 | Cropland, rainfed / Herbaceous cover | 10,129 | 90.79% | 80.11% | 10,157 | 83.01% | 62.56% |
| 3 | Cropland, rainfed / Tree or shrub cover | 12,127 | 99.54% | 99.20% | 11,336 | 99.07% | 98.42% |
| 4 | Cropland, irrigated or post-flooding | 5,162 | 99.99% | 99.91% | 5,217 | 99.93% | 99.81% |
| 5 | Mosaic cropland (>50%) / natural vegetation (tree, shrub, herbaceous cover) (<50%) | 4,896 | 95.70% | 96.12% | 5,439 | 93.14% | 93.47% |
| 6 | Mosaic natural vegetation (tree, shrub, herbaceous cover) (>50%) / cropland (<50%) | 25,702 | 91.44% | 89.54% | 27,613 | 82.67% | 79.58% |
| 7 | Tree cover, broadleaved, | 7,466 | 99.14% | 94.58% | 12,210 | 79.45% | 64.53% |

| | | | | | | |
|---|---|---|---|---|---|---|
| | evergreen, Closed to open (>15%) | | | | | |
| 8 | Tree cover, broadleaved, deciduous, Closed to open (>15%) | 12,658 | 94.14% | 93.40% | 12,094 | 94.76% | 93.53% |
| 9 | Tree cover, broadleaved, deciduous, Closed (>40%) | 15,445 | 90.60% | 91.34% | 14,557 | 91.21% | 90.13% |
| 10 | Tree cover, broadleaved, deciduous, Open (15-40%) | 198 | 100.00% | 100.00% | 198 | 100.00% | 100.00% |
| 11 | Tree cover, needleleaved, evergreen, closed to open (>15%) | 898,729 | 81.77% | 82.81% | 996,959 | 76.49% | 76.84% |
| 12 | Tree cover, mixed leaf type (broadleaved and needleleaved) | 10,529 | 83.71% | 83.61% | 12,351 | 79.36% | 84.00% |
| 13 | Mosaic tree and shrub (>50%) / herbaceous cover (<50%) | 22,384 | 90.10% | 90.79% | 24,002 | 85.67% | 90.28% |
| 14 | Mosaic herbaceous cover (>50%) / tree and shrub (<50%) | 5,159 | 90.48% | 91.17% | 4,955 | 92.15% | 93.02% |
| 15 | Shrubland | 217,565 | 89.63% | 89.83% | 236,376 | 85.49% | 85.47% |
| 16 | Grassland | 596,806 | 87.98% | 88.41% | 626,309 | 84.49% | 84.47% |
| 17 | Lichens and mosses | 2,711 | 92.62% | 92.88% | 3,153 | 86.24% | 93.78% |
| 18 | Sparse vegetation (tree, shrub, herbaceous cover) (<15%) | 25,240 | 88.56% | 88.26% | 28,065 | 83.54% | 87.57% |
| 19 | Shrub or herbaceous cover, flooded, fresh/saline/brakish water | 16,913 | 83.87% | 85.76% | 19,753 | 80.89% | 85.64% |
| 20 | Urban areas | 41,406 | 92.64% | 92.07% | 37,230 | 89.29% | 89.39% |
| 21 | Bare areas | 2,955 | 93.10% | 93.49% | 3,178 | 89.26% | 90.64% |
| 22 | Consolidated bare areas | 642 | 85.41% | 85.31% | 665 | 67.48% | 65.08% |
| 23 | Water | 5,213 | 81.31% | 81.96% | 6,120 | 73.57% | 74.23% |
| 24 | Permanent snow and ice | 0 | / | / | 0 | / | / |